# Library adaptors with integrated reference controls improve the accuracy and reliability of nanopore sequencing

Helen M. Gunter[1], Scott E. Youlten [2], Bindu Swapna Madala[2], Andre L. M. Reis[2], Igor Stevanovski [2], Ted Wong[2], Sarah K. Kummerfield[2,3], Ira W. Deveson [2,3], Nadia S. Santini[4], Esteban Marcellin [1] & Tim R. Mercer[1,2] ✉

Library adaptors are short oligonucleotides that are attached to RNA and DNA samples in preparation for next-generation sequencing (NGS). Adaptors can also include additional functional elements, such as sample indexes and unique molecular identifiers, to improve library analysis. Here, we describe Control Library Adaptors, termed CAPTORs, that measure the accuracy and reliability of NGS. CAPTORs can be integrated within the library preparation of RNA and DNA samples, and their encoded information is retrieved during sequencing. We show how CAPTORs can measure the accuracy of nanopore sequencing, evaluate the quantitative performance of metagenomic and RNA sequencing, and improve normalisation between samples. CAPTORs can also be customised for clinical diagnoses, correcting systematic sequencing errors and improving the diagnosis of pathogenic *BRCA1/2* variants in breast cancer. CAPTORs are a simple and effective method to increase the accuracy and reliability of NGS, enabling comparisons between samples, reagents and laboratories, and supporting the use of nanopore sequencing for clinical diagnosis.

Library adaptors are oligonucleotides that are attached to sample DNA fragments during the preparation of libraries for next-generation sequencing (NGS). Adaptors are an essential component of NGS workflows and are used in all library preparation protocols, including for short- and long-read sequencing, as well as DNA and RNA sequencing.

Library adaptors encode sequence elements, such as primer- and flowcell-binding sites, that are required for library preparation and sequencing[1]. They can also include additional sequence elements that confer additional functions, such as index barcodes that enable multiple libraries to be multiplexed and sequenced together in a single sequencing run. Unique molecular identifiers enable consensus error-correction strategies and can mitigate duplication artefacts resulting from the PCR amplification of low input samples[2–5].

Reference standards constitute ground-truth materials commonly used to measure the accuracy and performance of DNA and RNA sequencing experiments[6–11]. Natural reference materials, such as the NA12878 sample, are widely used as genomic controls but cannot be used as internal controls for individual samples[12]. Spike-in controls can be directly added to a sample prior to library preparation and act as internal controls[8,13,14]. However, their addition requires another step in the protocol and risks that an excess of spike-in control will be added and sequenced at the expense of the accompanying sample, which is particularly problematic for low input or degraded samples[15].

[1]Australian Institute of Bioengineering and Nanotechnology, University of Queensland, Brisbane, QLD, Australia. [2]Kinghorn Centre for Clinical Genomics, Garvan Institute of Medical Research, Sydney, NSW, Australia. [3]St Vincent's Clinical School, Faculty of Medicine, University of New South Wales, Sydney, NSW, Australia. [4]Programa Investigadoras e Investigadores por México, Instituto de Ecología, Universidad Nacional Autónoma de México, Mexico City, Mexico. ✉e-mail: t.mercer@uq.edu.au

To address these challenges, we developed CAPTORs (control adaptors), which are a class of library adaptors. They encode reference control sequences that measure qualitative and quantitative sequencing performance. CAPTORs can be used within any library preparation protocol, and their encoded information is retrieved and analysed during sequencing. Analysis of CAPTORs during nanopore sequencing provides a per-read measure of sequencing accuracy and quantitative library bias. These analyses can benchmark sequencing performance, enable improved normalisation between multiple libraries, and correct for sequencing errors during the diagnosis of mutations in cancer genes. Together, we provide CAPTORs as a simple and effective approach that seamlessly incorporates qualitative and quantitative reference controls into the library preparation workflow to improve the accuracy and reliability of sequencing.

## Results

### Design of CAPTORs (control library adaptors)

We first designed synthetic, custom adaptors for use in Oxford Nanopore Technologies (ONT) sequencing (Fig. 1a). The long reads generated by ONT sequencing permit the use of longer adaptors with a greater range of informational content than is otherwise possible with short-read sequencing. We designed 72 adaptors, each with a length of 90 nucleotides (nt) (Fig. 1b). Each adaptor was designed to include three regions: (i) a leading 5′ 30 nt constant sequence that is identical for all CAPTORs and acts as a 'burn-in' region; (ii) a central, variable 30 nt region that differs between each CAPTOR, which collectively represent a diversity of 6-mers that can be used to evaluate ONT base-calling accuracy; (iii) a final 3′ constant sequence that is identical in all CAPTORs to prevent preferential ligation during library preparation.

We manufactured the CAPTORs using enzymatic DNA synthesis using the DNA Script SYNTAX instrument (see Methods). The CAPTORs were then pooled into a master mix and used as adaptors during standard ligation library preparation (Fig. 1a and Supplementary Fig. S1). Prepared libraries were then sequenced on an ONT MinION instrument (see Methods). The output read files were then analysed, with each terminal CAPTOR sequence identified and classified by its unique variable sequence. We then evaluated sequencing accuracy in the variable region by comparing each read sequence to its corresponding ground-truth reference sequence (Fig. 1c and Supplementary Fig. S2a, b).

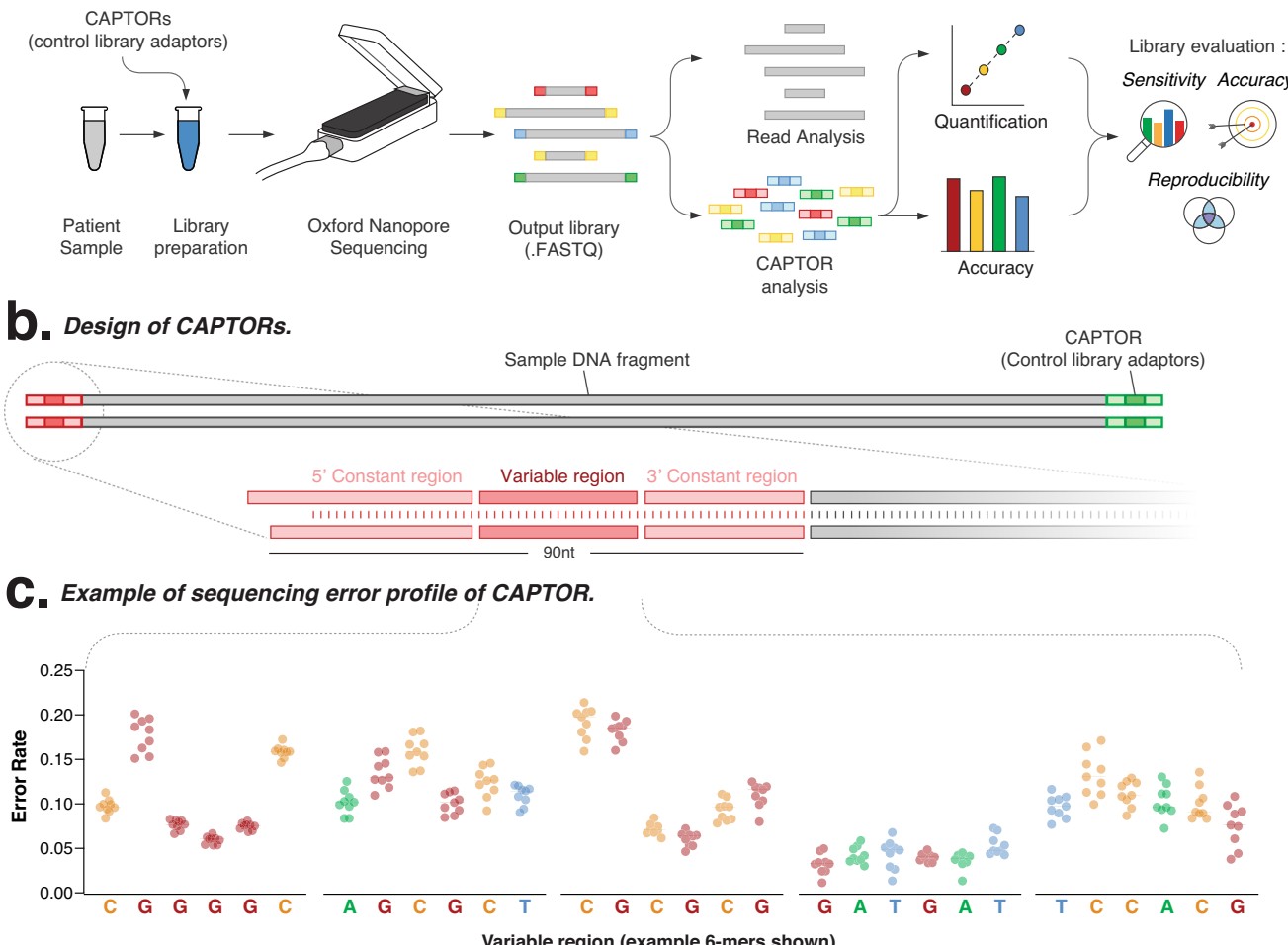

**Fig. 1 | Schematic overview of design and use of CAPTORs. a** Schematic overview of workflow showing the integration of CAPTORs within the library preparation step of the Oxford Nanopore Technologies (ONT) sequencing workflow. Analysis of CAPTORs in output libraries enables qualitative and quantitative evaluation of library performance. **b** Schematic diagram shows ligation of CAPTORs to the 3′ and 5′ termini of sample DNA fragments, with detailed inset showing CAPTOR design (including 5′ constant region, middle variable region, and 3′ constant region). The variable region differs between the CAPTORs, and collectively encompasses a diversity of 6-mer sequences. **c** Histogram shows the per-nucleotide sequencing error rate for an example CAPTOR across replicate libraries (*n* = 9). Source data are provided as a Source Data file.

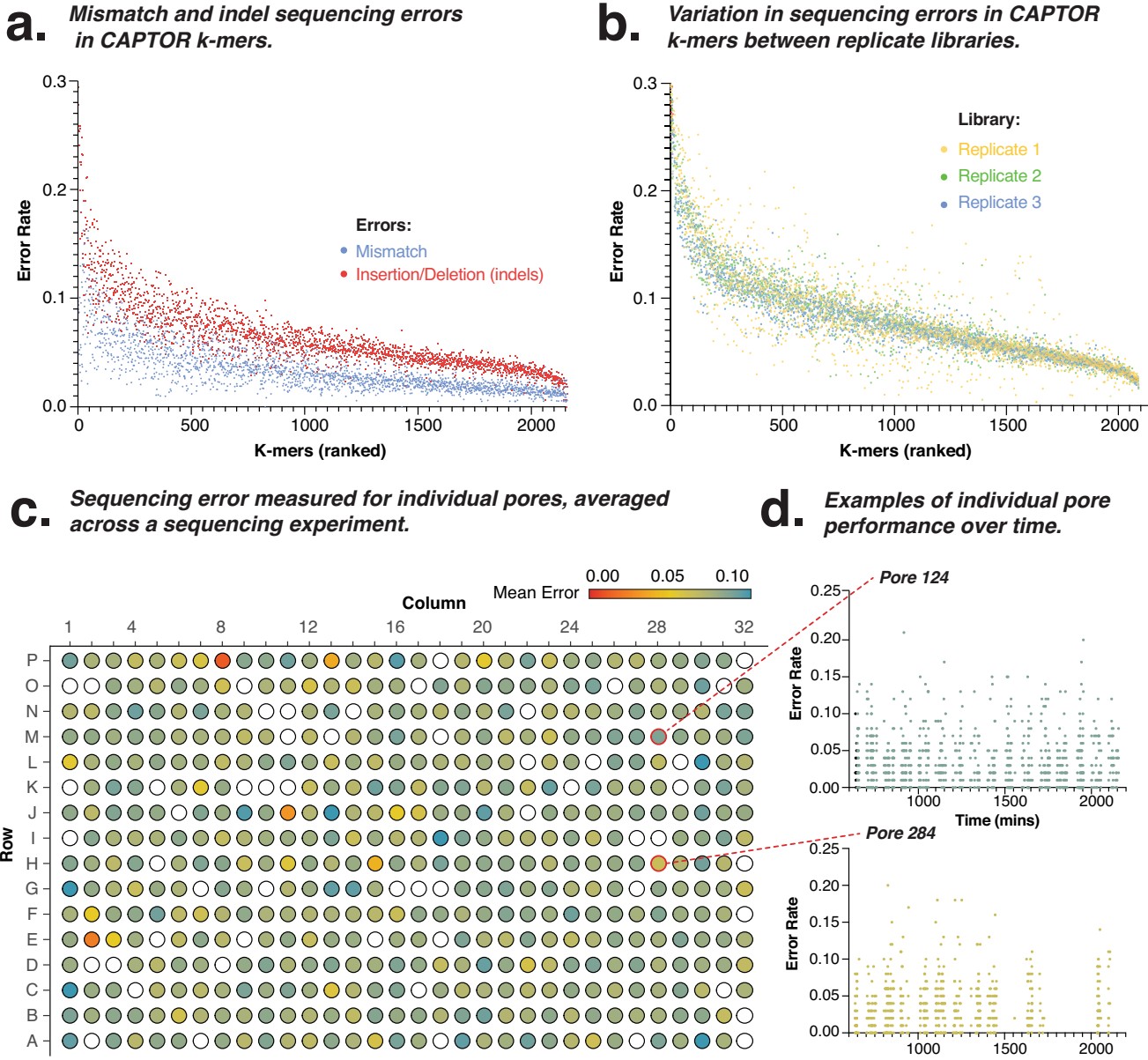

**Fig. 2 | Measuring sequencing accuracy using CAPTORs. a** Scatter plot shows the mismatch and indel sequencing error profiles for 6-mers (ranked by total error rate) in CAPTORs from a single library. **b** Scatter plot shows the sequencing error profiles for 6-mers (ranked by mean error rate) in replicate libraries (*n* = 3). **c** Schematic diagram shows an Oxford Nanopores Technology (ONT) flowcell matrix with individual pores coloured according to mean sequencing error across the duration of an experiment. **d** Scatter plots show sequencing error profiles for example pores across the duration of an experiment. Source data are provided as a Source Data file.

## Using CAPTORs to benchmark sequencing accuracy

We initially used CAPTORs to prepare a library from synthetic, mock microbial communities using the LSK109 protocol (see Methods). This mock community comprises synthetic microbial genomes that provide a useful reference sample to validate the performance of CAPTORs[16].

We first measured the sequencing accuracy of all 6-mers represented within the variable regions of the CAPTORs (Fig. 1c). This provided a detailed, complex and comprehensive profile of sequencing errors for the individual library (Fig. 2a). We observed a mean per-base error rate (mean = 0.089; SD = 0.035) similar to previously reported error rates for MinION sequencing[17]. This total error rate included differing contributions of mismatch (mean error = 0.034; SD = 0.021), insertion and deletion (indel) errors (mean error = 0.062; SD = 0.033, Fig. 2a and Supplementary Fig. S2c). Wide variation (7-fold) was also

observed between the most- and least-accurate 6-mers (AATCGA, 0.030 errors/nt and CGGGGG, 0.219 errors/nt, respectively).

Next, we investigated the factors that influence the sequencing error rate among k-mers. Error rates were greatest for repetitive and low-complexity k-mers, a known source of error for ONT base callers (Supplementary Fig. S3a)[18]. Errors at repeats are also progressive, with the error rate increasing in proportion to the repeat length (Supplementary Fig. S3a). We also observed a GC bias in sequencing accuracy, with a higher error rate for 6-mers with high GC compared to low GC content (Supplementary Fig. S3b). Moreover, the contribution of GC and repeat bias was cumulative, with the highest error rate observed for GC-rich homopolymer k-mers (Supplementary Fig. S3a).

To investigate whether these errors are derived from random or systematic variation, we compared CAPTOR sequencing error profiles across replicate libraries (Fig. 2b and Supplementary Fig. S3c).

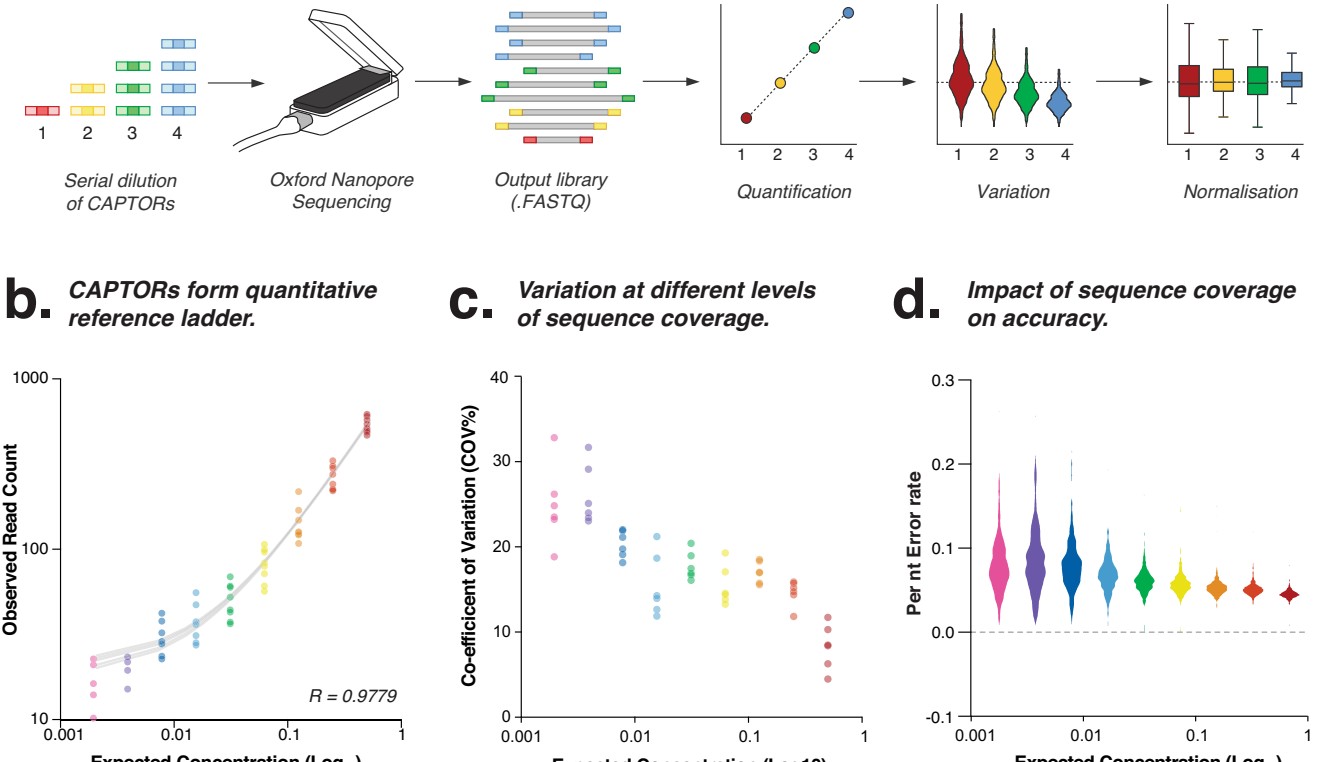

**a.** *Use of CAPTORs for quantitative analysis of nanopore sequencing.*

Serial dilution of CAPTORs · Oxford Nanopore Sequencing · Output library (.FASTQ) · Quantification · Variation · Normalisation

**b.** *CAPTORs form quantitative reference ladder.*

R = 0.9779

**c.** *Variation at different levels of sequence coverage.*

**d.** *Impact of sequence coverage on accuracy.*

**Fig. 3 | Quantitative analysis using CAPTORs. a** Schematic diagram illustrates the preparation of a master mixture that constitutes a staggered dilution of CAPTORs. Use of the staggered CAPTORs master mix generates an internal quantitative reference ladder that can be used to measure technical variation and improve normalisation between samples. **b** Scatter plot compares the observed CAPTOR count relative to their expected concentrations, forming a staggered reference ladder to evaluate the quantitative performance of an individual library. 95% confidence intervals of line of best fit are indicated. **c** Scatter plot compares the quantitative reference ladder formed by the CAPTORs relative to the read counts for synthetic microbes within a mock microbial community sample. **d** Violin plot shows the sequencing error for CAPTORs at different concentrations, illustrating the impact of sequencing coverage on accuracy. Source data are provided as a Source Data file.

Comparison of k-mer sequencing accuracy showed little variation between technical replicates (mean 8.4% difference between replicate k-mer sequence error rates; Supplementary Fig. S4a, b). This reproducibility of errors was greater for insertion and deletion errors between libraries (mean 6.7% difference) than for mismatch errors (mean 12.1% difference; Supplementary Fig. S4c–e). This high reproducibility of errors indicates they are primarily derived from systematic rather than random sources and may be modelled and normalised to improve sequencing accuracy (see below)[19].

**Measuring individual pore performance using CAPTORs**

The retrieval and analysis of CAPTOR information during sequencing allows for the ongoing measurement of read, pore and flowcell performance. CAPTORs are the first region of the read to traverse the nanopore and be sequenced, thereby providing an early measure of sequencing accuracy for individual reads. We found that mean CAPTOR sequencing accuracy matches the mean sequencing accuracy of the adjacent microbial DNA sequence (Supplementary Fig. S5a). Similarly, we found the sequencing error rates of CAPTORs for 'failed' reads (median error rate = 0.068) was greater than for 'passed' reads (median error rate = 0.045, p value < 0.0001, Supplementary Fig. S5b). This initial measure of CAPTOR accuracy may be incorporated within adaptive sequencing strategies to provide an early evaluation of the sequencing performance of individual reads or pores[20].

We next used CAPTORs to measure variability in individual pore performance, with sequencing accuracy of pores varying on average 3.2-fold across the duration of the experiment, with poorly performing, inaccurate pores also having low sequencing throughput (Fig. 2c, d and Supplementary Fig. S5c, d). Although we observed fluctuating error rates for each pore across the duration of the experiment, we did not observe any significant temporal trends (Supplementary Fig. S5e, two-way ANOVA p = 0.1308, for pores that remained active throughout the 72 h sequencing period). The position of a pore on the flowcell also had no apparent impact, with the performance of individual pores independent of other pores (Fig. 2c).

CAPTORs can also benchmark the performance of different sequencing reagents and methods. We used CAPTORs to evaluate the sequencing accuracy of different nanopore versions. The R10.3 nanopore, which has a longer barrel and a dual reader head, has been developed to enhance the accuracy of homopolymer regions[21]. We used matched CAPTOR libraries to compare the error profile of the R10.3 pore to R9.4.1 pore performance. As expected, the R10.3 pore exhibited a distinct error profile, with a lower mean error rate (0.037 error/nt) compared to the R9.4.1 pore (0.045 error/nt), which is largely due to the lower insertion rate for the R10.3 pore (0.021 error/nt, compared to the 0.032 error/nt for R9.4.1, Supplementary Fig. S6a, b). This distinction in R10.3 pore performance, as measured by CAPTORs, is most notable at low-complexity repeats (R10.3 mean

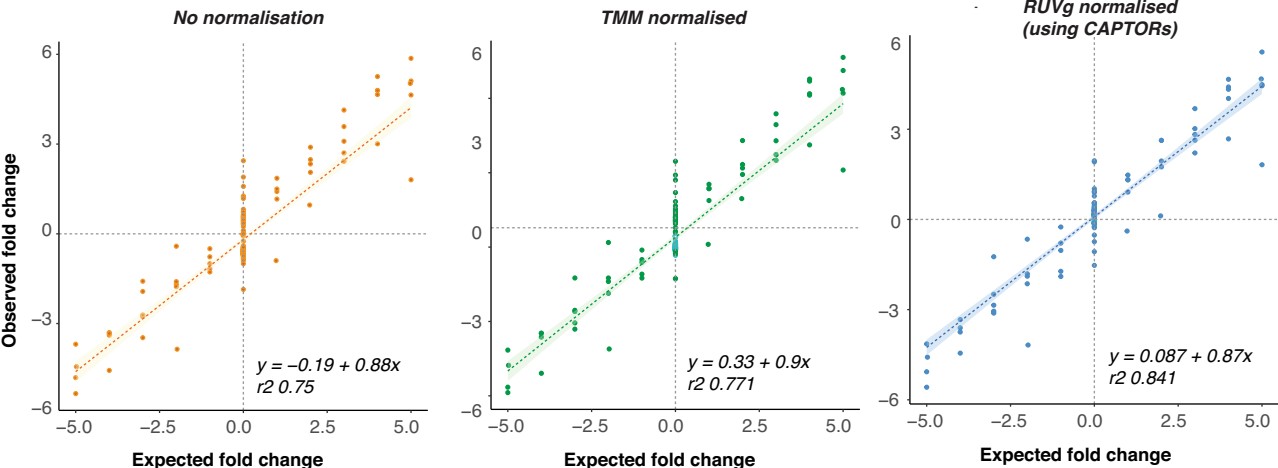

**a.** *Detection of log-fold changes in microbe abundance.*

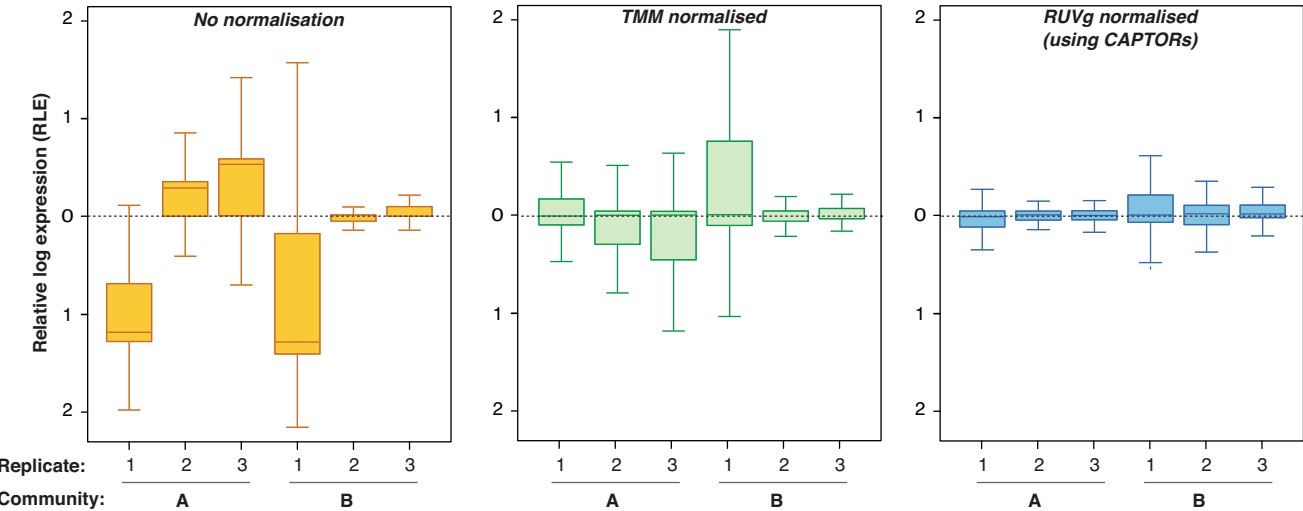

**b.** *Relative log expression (RLE) plots show unwanted variation in microbe abundance.*

**Fig. 4 | Measuring fold-change differences between samples using quantitative CAPTORs. a** Scatter plot compares the expected and observed fold-changes between mock microbial communities with no normalisation, Trimmed Mean of M-values (TMM) normalisation and removal of unwanted variation (RUVg) normalisation using CAPTORs. 95% confidence intervals of line of best fit are indicated for each plot. **b** Relative log expression (RLE) plots show unwanted variation within replicate mock microbial communities following no normalisation, TMM normalisation and RUVg normalisation using CAPTORs. Data derived from three technical replicates, prepared in separate laboratories. Source data are provided as a Source Data file. Quartiles of box plots are indicated (25%, 50% and 75%).

error = 0.048, R9.4.1 mean error = 0.083, Supplementary Fig. S6c). These empirically determined sequencing error rates differ from manufacturer's reports[21] and demonstrate how CAPTORs can measure the sequencing performance of each library, benchmark new chemistries and base-calling algorithms and inform best-practise guidelines to optimise sequencing performance.

### Using CAPTORs to measure quantitative accuracy
Sequencing can measure quantitative features within a sample, such as gene expression, copy-number variation and microbial abundance. Given that CAPTORs are ligated in a constant ratio to the accompanying sample DNA fragments, the quantitative performance of the CAPTORs directly matches the quantitative performance of the accompanying DNA sample. Therefore, we next used CAPTORs as internal quantitative reference controls to measure the sensitivity and complexity of nanopore libraries.

We first prepared a master mixture of CAPTORs, wherein each CAPTOR is titrated at two-fold serial dilutions, which are then combined into a single master mixture (Supplementary Fig. S1 and Fig. 3a). This CAPTOR master mixture was then used to prepare libraries from mock microbial communities for ONT sequencing (as described above). The variable CAPTOR sequences were then retrieved from each read, counted and compared to the expected CAPTOR concentration to generate a staggered reference ladder that can measure quantitative library features[22] (see Methods).

To demonstrate this approach, we compared observed counts for individual CAPTORs to their expected relative concentrations, thereby generating a quantitative reference ladder associated with each library (Fig. 3b). This ladder indicates the overall quantitative accuracy of the library ($R^2 = 0.9779$) and the uncertainty associated with quantitative measurements of differing abundance, at different read depths, in different samples (Fig. 3c, d and Supplementary Fig. S7a–c).

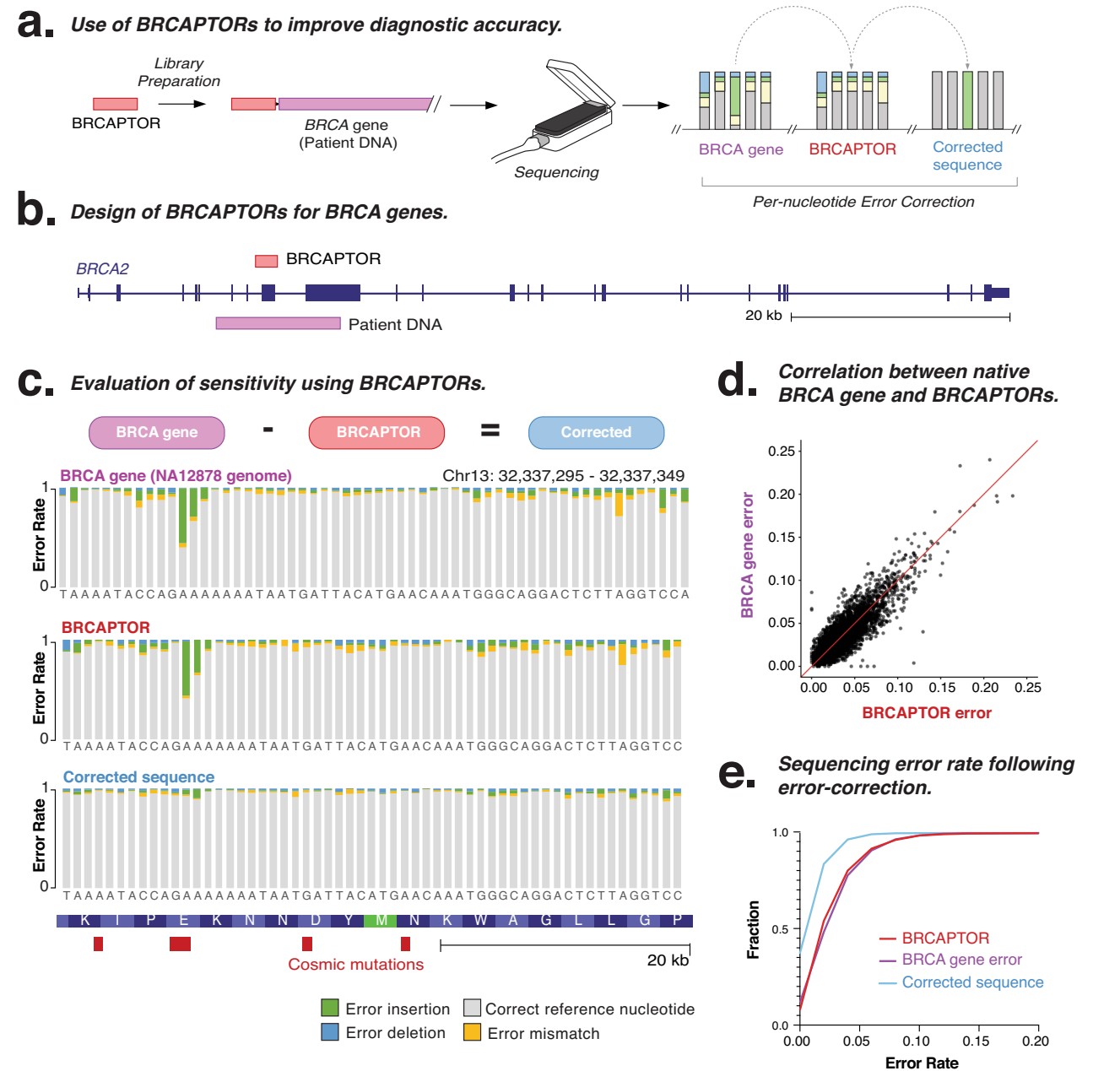

**Fig. 5 | Improving the diagnosis of mutations in *BRCA* genes using BRCAPTORs.**
**a** Schematic diagram illustrates the use of CAPTORs that encode reference control sequences for clinically important genes to perform per-nucleotide error correction of accompanying human patient DNA samples. **b** Genome browser shows the design of BRCAPTORs representing a clinically important sequence from the *BRCA2* gene. **c** Per-nucleotide error profile shows the sequencing error rate for corresponding gene sequences within NA12878 human DNA sequence, BRCAPTOR, and error-corrected sequence. **d** Scatter plot indicates the correlation between sequencing errors in NA12878 human DNA relative to BRCAPTORs. **e** Cumulative frequency distribution shows the sequencing error profile of NA12878 human DNA sequence, BRCAPTOR, and error-corrected sequence. Source data are provided as a Source Data file.

The limit of detection and limit of quantification (LOQ) are key metrics used within clinical laboratories to describe diagnostic performance[23]. To demonstrate how we can determine these metrics from CAPTORs, we subsampled the library to different read depths (Supplementary Fig. S7d). We then tested each library to determine the minimum read depth required to achieve reliable quantification of CAPTORs. We found a minimum sequencing coverage of $\sim 5 \times 10^4$ reads, which was achieved during the first ~2 h of sequencing, which was required to achieve reliable quantification across the full dynamic range of CAPTORs (to <1% frequency; Supplementary Fig. S7d). Below this threshold, we observed increasing quantitative uncertainty illustrated by a wide confidence interval at lower sequencing depths

(Fig. 3d). This demonstrates how ongoing real-time analysis of the CAPTORs could be used to ensure minimal sequencing thresholds are attained according to the desired level of accuracy and sensitivity. This minimum threshold may vary between experiments, and will depend on several factors, including the experimental aims, desired sensitivity, and the particular analysis being performed[24–26].

**Normalisation of metagenome samples with CAPTORs**
Unwanted technical variation introduced during library preparation and sequencing can confound comparisons between samples and prevent the reliable detection of fold-change differences. To investigate variation between libraries, we used CAPTORs to prepare six

replicate libraries from two distinct mock microbial communities with known fold-change differences in synthetic microbial abundance[16]. We first measured CAPTOR ladders, finding high reproducibility across replicate libraries (mean 1.7% difference between replicate quantitative measurements; Supplementary Fig. S7a). We also measured the abundances of the accompanying synthetic microbes, and compared the observed and expected fold-change differences in synthetic microbial abundance between the two mock communities (Fig. 4a).

Given the ability of CAPTORs to measure quantitative technical variation, we next investigated whether CAPTORs could be used as constant scaling factors to mitigate batch-effect differences between libraries. In this case, CAPTORs were used as negative scaling factors with the removal of unwanted variation (RUVg) normalisation method designed to compare samples according to shared spike-in controls[27]. An advantage of using CAPTORs as scaling factors with RUVg normalisation is the ability to relax the common normalisation assumption that most microbes are not differentially abundant between samples[28].

To benchmark the use of CAPTORs during normalisation, we compared RUVg (with CAPTORs) to alternative current best-practice methods, such as Trimmed Mean of M-values (TMM) normalisation (Fig. 4b)[29]. We evaluated performance according to the true-positive and true-negative detection of known fold-change differences between microbial communities, finding that RUVg normalisation with CAPTORs outperformed TMM, and improved the detection of known fold-change differences in synthetic microbe abundance between the two mock communities (Supplementary Fig. S8a). This demonstrates how samples prepared using a common CAPTOR master mix can effectively normalise unwanted technical variation between libraries and improve the detection of bonafide fold-change differences.

### Using CAPTORs in RNA sequencing

RNA sequencing can provide a global transcriptome profile and is a primary tool used in gene expression analysis. To investigate the incorporation of CAPTORs within the RNA sequencing workflow, we used CAPTORs to prepare cDNA libraries from Universal Human Reference RNA (UHRR), a reference RNA sample selected because its expression profile has been well characterised[30–34].

We first analysed the quantification of CAPTORs within the RNA sequencing libraries, indicating library sensitivity and quantitative accuracy (Supplementary Fig. S8b). The measured abundance of CAPTORs was plotted against relative input concentration, revealing a strong linear trend ($R^2 = 0.9552$) to a lower inflection point determined using segmental linear regression analysis, occurring at ~7.1 reads. This indicated the LOQ[23] below which the measurement of CAPTOR abundance becomes more variable ($R^2 = 0.2210$). It should be noted that, unlike conventional spike-ins, CAPTORs are in constant proportion to the accompanying samples due to their direct incorporation into each sequenced read.

The UHRR sample includes many expressed genes that span a wide range of expression levels. We measured the expression of these human genes and compared this to the reference ladder formed by the CAPTORs (Supplementary Fig. S8c). We found 5903 genes (comprising the top 36.1% of the 16,354 GENCODE genes detected) in the accompanying UHRR sample exceeded the LOQ and may be considered sufficiently sampled for accurate gene expression measurements within this library (Supplementary Fig. S8c). In addition, the CAPTORs can also estimate the uncertainty associated with the measurement of specific genes. This demonstrates how CAPTORs can be used routinely to provide an empirical measure of confidence in gene expression profiling with RNA sequencing, even within a single library.

### Improving cancer diagnosis with CAPTORs

Sequencing has become increasingly used in oncology, where it can identify somatic mutations that cause cancer[35]. However, somatic mutations are often present at low frequency, and their reliable diagnosis can be confounded by the inaccuracies of ONT sequencing.

Numerous error-correction tools have been developed to model ONT sequencing errors and improve its accuracy[36]. These tools often employ a range of machine learning and homology-based methods to model and mitigate systematic errors[19,37,38]. Given their ability to measure sequencing error, we next considered whether CAPTORs could be similarly used as integrated reference controls to empirically model the sequencing error profiles of clinically important genes and thereby assist in the interpretation and error correction of ONT data (Fig. 5a).

To demonstrate this strategy, we designed custom *BRCA* CAPTORs (termed BRCAPTORs) that encode synthetic versions of the *BRCA1* and *BRCA2* gene sequences. Both *BRCA* genes are major susceptibility loci for breast cancer. They include repetitive sequences that are susceptible to insertions or deletions that cause frameshift loss-of-function mutations, thereby representing strong candidates for the development of reference controls[39–41].

We designed three BRCAPTORs that encode reference sequences for one exon within *BRCA1* (5172 nt) and two exons within *BRCA2* (2054 and 2254 nt, Fig. 5b and Supplementary Fig. S9a). The BRCAPTORs were used to prepare libraries from natural *BRCA1* and *BRCA2* gene sequences from the NA12878 human genome DNA sample[42]. This resulted in *BRCA1* and *BRCA2* genomic DNA fragments attached by flanking BRCAPTORs that provide ground-truth sequences to establish a background sequencing error profile for the accompanying human *BRCA* genes.

We compared the sequencing accuracy of the BRCAPTORs with the attached NA12878 human *BRCA* genes, showing correlated error profiles for mismatches, insertions and deletions (Fig. 5c, d and Supplementary Fig. S9b–d). Given this concordance, we used the BRCAPTOR error profile to perform nucleotide-by-nucleotide normalisation of the accompanying human *BRCA1/2* gene error profiles (Fig. 5c and Supplementary Fig. S10a). Using this approach, we reduced the median error rate in the error-corrected patient DNA sequence from 0.042 to 0.018 (Fig. 5e). We found this per-nucleotide error-correction approach was most effective for deletion errors, which show the strongest degree of systematic error, where the mean error rate was reduced from 0.020 to 0.007 (Supplementary Fig. S9c, d). To determine whether this error-correction strategy could improve the diagnosis of clinically relevant *BRCA* mutations, we focused on mutations listed in COSMIC[43], finding the median error rate was reduced from 0.032 to 0.012 for these cancer-associated mutations (Supplementary Fig. S10b). This proof-of-principle experiment demonstrates how CAPTORs containing clinically important sequences can provide internal controls to guide error-correction tools and improve the interpretation and accuracy of ONT sequencing data during clinical diagnosis[36]. However, while this approach can include genes of diagnostic importance, it is limited to smaller gene panels, and standard spike-ins may be more suitable for representing larger numbers of genes.

## Discussion

Reference standards are needed to understand the sequencing accuracy and quantitative performance of NGS libraries. Currently available reference standards include both natural reference genome materials (such as the NA12878 genome) and synthetic spike-in controls (such as sequins, ERCC and SIRV controls)[6,11,14,16,42,44]. Although synthetic spike-ins have the advantage of measuring internal library variation, they must be precisely added to a sample during library preparation, must be bioinformatically calibrated, and risk overwhelming low input or degraded samples.

Here we describe the design and validation of a class of library adaptors, termed CAPTORs, that incorporate qualitative and quantitative reference controls. CAPTORs confer many of the benefits of

reference standards but can be routinely incorporated into library preparation reagents during the NGS workflow. Like other reference standards, CAPTORs can measure sequencing performance and quality control, enable rapid troubleshooting, and benchmark different methods, reagents or instruments. Routine use of CAPTORs, which can be seamlessly incorporated into the NGS workflow, will measure performance and inform operational decisions. We show how CAPTORs can distinguish the sequencing error profiles of different libraries, measure individual read or pore performance across the duration of the sequencing experiment, and benchmark protocols, reagents or methods.

The CAPTORs can incorporate diverse k-mers or specific gene sequences of interest (that cannot be otherwise determined from standard library adaptors). Given that CAPTORS are the first part of the read to traverse the nanopore channel and be sequenced, they can provide an immediate measure of sequencing performance. This responsive analysis can be incorporated within 'CAPTOR-aware' adaptive sequencing strategies to provide real-time evaluation of library accuracy and complexity[20].

CAPTORs can determine the sensitivity, quantitative accuracy and bias of NGS libraries. These quantitative metrics are needed to measure gene expression in RNA sequencing, microbe abundance in metagenomics or copy-number variation and heterozygosity in genomics. Combining different CAPTORs at different concentrations into a master mix can provide an internal, staggered reference ladder within each library. Furthermore, CAPTORs are ligated to the termini of DNA fragments at a constant ratio, ensuring their quantitative counts and dynamic range are directly proportional to the accompanying sample. As a result, the CAPTORs can directly measure the quantitative accuracy and complexity of a library and confirm whether sufficient sequencing depth has been achieved according to the desired sensitivity and confidence[26].

Given this ability to measure quantitative bias and technical variation within a library, CAPTORs can also normalise technical differences between samples[45]. We showed that normalisation using CAPTORs (in conjunction with RUVg[27]) resulted in improved detection of known fold-change differences in comparison to current best-practise normalisation models[27]. As a result, any libraries prepared using a shared CAPTOR master mix can be normalised using our best-practice technique, enabling more accurate comparisons and interoperability between libraries. This is particularly useful for normalisation across large patient cohorts, longitudinal patient timelines, and laboratories.

Numerous read polishing and error-correction tools have been developed to model and mitigate sequencing errors in ONT data[19,36]. We provide a proof-of-principle demonstration that CAPTORs can be similarly used to empirically model the background sequencing error of clinically important gene sequences and assist in the per-nucleotide error correction and interpretation of ONT datasets. We show how the use of CAPTORs designed to represent *BRCA* genes improves the accuracy of nanopore sequencing, which remains a key challenge in the adoption of ONT sequencing in clinical diagnosis. Although the design of gene-specific CAPTORs is not practical for all genes, this approach is suitable for small panels of selected genes with high diagnostic importance and complex error profiles.

Within this study, we designed and synthesised CAPTORs for use with nanopore sequencing, whose long-read and error profile benefits from CAPTORs. However, CAPTORs could also potentially be used with other sequencing platforms such as short-read Illumina sequencing. Due to the short read length, the control elements would necessarily be short (we suggest 12 nt, in comparison to the 90 nt used for nanopore CAPTORs) and would not encode extended reference sequences, required to provide a comprehensive analysis of sequencing accuracy. In addition, the control elements would also need to be sufficiently diverse to ensure optimum cluster discrimination at each sequencing cycle. Nevertheless, the CAPTORs could feasibly provide

quantitative reference ladders that measure the sensitivity and quantitative accuracy of short-read sequencing libraries. Short-read CAPTORs could be combined in a dilution series, permitting the quantitative scaling of metagenomics and RNA-seq libraries, using the approach demonstrated for nanopore sequencing. Furthermore, barcoded adaptors, which are widely used in single-cell and spatial transcriptome sequencing methods, can similarly incorporate quantitative reference control sequences and confer the benefits of CAPTORs to measure single-cell library complexity and inform normalisation between individual cells.

Reference controls are a central requirement for ensuring the accuracy and reliability of sequencing technologies for clinical diagnosis. The incorporation of reference controls within library adaptors, as demonstrated here with CAPTORs, ensures these benefits are seamlessly integrated within libraries without requiring any additional steps. As a result, we propose the routine use of CAPTORs, which will allow laboratories to monitor sequencing performance, benchmark new technologies and ensure the reproducibility of NGS results.

## Methods
### Design of CAPTORs
We designed 72 unique 90 nt adaptors, termed CAPTORs, with the following structure. (i) A 30 nt 5' region with an invariable sequence, included as a 'burn-in' region. This is due to the high error rate that is typical of ONT sequencing in the first 15–20 nt of each sequence. This sequence was chosen from randomly generated sequences that had been previously found to perform accurately and consistently during ONT sequencing[16]. (ii) A central 30 nt region that was unique to each of the 72 CAPTORs. The central variable region was designed based on a sequence containing all possible 6-mers generated using Shortcake software[36]. CAPTOR sequences were analysed using BLAST (Nucleotide Collection nr/nt; Megablast, 1–2 Match Mismatch Score, Linear Gap Costs) to ensure they did not exhibit extended (>20 nt) homology to natural sequences. CAPTOR sequences were analysed with the Predict a Secondary Structure Web Server[46] to ensure there were no extended (>8 nt) hairpin structures.

### Synthesis of CAPTORs
CAPTOR adaptors were synthesised by enzymatic DNA synthesis using a DNA Script SYNTAX System. The oligos were desalted automatically on the system and were eluted in nuclease-free molecular biology-grade water. They were quantified using the system's onboard spectrophotometer that measured UV absorption at 260 nm and was normalised by the system to a final concentration of 2 μM. The CAPTORs were pooled to form a staggered ladder (Supplementary Fig. S1).

### Oxford Nanopore Sequencing with CAPTORs
Libraries were prepared from DNA samples (see below) using the LSK109 Ligation Sequencing protocol, according to the manufacturer's protocols (Oxford Nanopore Technology). Briefly, 1 mg of each sample was sheared into 25 kB fragments, using Covaris g-tubes. Each library was loaded onto a separate R9.4.1 or R10.3 flowcell and was sequenced on a GridION instrument for 72 h with live base-calling enabled (Guppy v4).

### Analysis of sequencing accuracy using CAPTORs
To analyse the sequencing accuracy of CAPTORs, we first determined the base-wise error rates for CAPTOR sequences in each sequencing library. Reads were clipped to the first 500 nt using fastp[47] and aligned to a custom reference index of CAPTOR sequences using MiniMap2 v2.17-r941 with the parameters 'minimap2 -ax map-ont' optimised for Oxford Nanopore libraries[48]. The resultant .SAM/BAM files were then sorted and indexed using samtools[49]. The per-nucleotide error profile relative to the reference CAPTOR sequence was determined using pysamstats[50].

The CAPTOR variable sequences were used to determine the sequencing error rate of 6-mers in each library. CAPTOR sequences and base-wise error statistics were subset to just the 30 nt variable regions of each adaptor in R (v4.0.2). Variable regions were classified into overlapping sliding 6-mer windows, with the sequencing error profile averaged across these windows and assigned to the corresponding 6-mer sequence using the extractList function of the IRanges R-package (v2.22.2). Where a 6-mer was present in more than one CAPTOR, the mean across all instances was used. Sequencing error rates for 6-mers with different sequence properties (i.e., GC or homopolymer content) were compared using Brown-Forsythe and Welch's ANOVA for unmatched data in GraphPad Prism (v9.0.0). CAPTOR sequences were classified according to.FASTQ header details. To evaluate per-read, per-pore and time-dependent analysis of sequencing error rate, BAM files were split into individual CAPTOR sequences using bamtools[51]. Error statistics were calculated across CAPTOR sequences for each read using pysamstats, with read, pore and time of sequencing extracted from the.FASTQ headers of each read.

### Preparation of quantitative CAPTOR mixtures

To generate a staggered serial dilution series, the 72 CAPTORs were first divided into groups of nine CAPTORs. Each CAPTOR group was then diluted across an 8-fold dilution series to generate a range of concentrations ranging from undiluted to 1:128 (Supplementary Fig. S1). The DNA concentrations in each dilution were then verified using the Qubit instrument (Invitrogen). Equal amounts of each dilution were then mixed to form a single master mix. The CAPTOR master mix was then used during standard library preparation and sequencing as described above. To analyse the staggered CAPTOR dilutions, the CAPTORs at the 5′ termini of sequenced reads were classified according to the variable sequences. The observed read count for each CAPTOR sequence was then compared to the expected dilution to assemble a staggered reference ladder. The impact of sequencing depth was evaluated via the bioinformatic subsampling of libraries to variable depths using the seqtk sample tool (version 1.0-r82-dirty). The quantitative analysis was then repeated for subsampled libraries as described above. Plotting and statistical analysis were performed using the GraphPad Prism v9.0.0 software.

### Metagenome experiment

ONT libraries were prepared in triplicate from Mixture A and Mixture B synthetic mock microbial communities[16] using the LSK109 library preparation protocol as described above. The resulting.FASTQ libraries were then aligned to the CAPTOR sequences described above and to metasequin sequences (from www.sequinstandards.com/resources). Read counts were calculated as the mean read depth aligned across each reference sequence. The observed read count for either the metasequins or CAPTORs was compared to the expected concentration. In addition, the observed fold differences between the metasequins in Mixture A and B were compared to the expected fold-change differences. The normalisation of replicate samples was performed using the TMM[52] using EdgeR (version 3.26.0)[53], or the RUVg[27]. Normalised read counts were then compared to the expected abundance of each synthetic microbial sequence, and the $p$ value significance of known fold-changes between Mixture A and B was determined. Plotting and statistical analysis were performed using the GraphPad Prism v9.0.0 software.

### RNA sequencing experiment

ONT libraries were prepared from UHRR, a reference RNA mixture generated from 10 different cell lines[19]. RNA was first converted to double-stranded cDNA using Superscript IV Reverse Transcriptase (ThermoFisher). CAPTORs were ligated to cDNA molecules, and the libraries were prepared using the ONT SQK-LSK109 kit as described

above. The resulting libraries were then sequenced on either R9.4.1 or R10.3 MinION flow cells.

### BRCAPTOR design and sequencing experiment

We designed custom BRCAPTORs that encode reference sequences for one exon within *BRCA1* (5172 nt) and two exons within *BRCA2* (2054 and 2254 nt, see Supplementary Data 1). Our BRCAPTOR pool included three custom adaptors that spanned the entire length of the selected *BRCA* exons. BRCAPTORs were manufactured and purified using a DNA Script SYNTAX System as described above.

*BRCA1* and *BRCA2* genes were amplified using Taq Polymerase from NA12878[29], a NIST reference sample. The resulting PCR products were then ligated to the custom BRCAPTORs using DNA ligase (New England Biolabs). The resulting combined fragments were then prepared and sequenced using a MinION instrument on an R9.4.1 flowcell as described above.

Output data (.FASTQ) were then analysed as follows. FASTQ libraries were first aligned to a custom reference index comprising the BRCAPTOR and *BRCA* sequences using MiniMap2[48]. This enabled BRCAPTOR and *BRCA* sequences to be distinguished according to their alignment to the reference index and their flanking orientation within each read. Partial length or aligning reads were omitted from further analysis. Resulting.SAM/BAM files were pre-processed using samtools[49]. The per-nucleotide error profile relative to the reference index sequence was determined using pysamstats[50]. To perform simple error correction, the per-nucleotide error profile of the BRCAPTOR sequences was subtracted from the corresponding nucleotides within the *BRCA* sequences. This analysis was also restricted to annotated pathogenic variants listed in the COSMIC database[43].

### Statistics and reproducibility

Triplicate samples were included in our metagenomics and CAPTOR analyses. No statistical method was used to determine this sample size. We selected this number of replicates as it reflects a common NGS experimental design, for which we aim to provide error corrections. The replicates were prepared in separate laboratories to demonstrate the technical errors that can arise during library preparation. No data were excluded from our analyses. Our experiments were not randomised. The investigators were not blinded to allocation during experiments and outcome assessment, as the preparation of shotgun sequencing libraries is unlikely to be impacted by prior knowledge of sample content. All bioinformatic analyses were performed centrally, to reduce any potential biases in data interpretation.

### Reporting summary

Further information on research design is available in the Nature Research Reporting Summary linked to this article.

## Data availability

All sequencing data generated in this study have been deposited in the Sequence Read Archive with the BioProject Accession Identifier PRJNA781348. ONT CAPTOR and BRCAPTOR sequences are also available in Supplementary Data 1. The COSMIC database used in this work is available via the following link: https://cancer.sanger.ac.uk/cosmic. Source Data are provided with this paper.

## Code availability

Scripts used for the analysis of CAPTORS can be accessed via https://github.com/mercertim/Captors.

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

## Acknowledgements

We acknowledge the following funding sources: National Health and Medical Research Council (NHMRC grants APP1108254, APP1114016, APP1136067 to T.R.M.), UNSW Tuition Fee Scholarship (TFS; to A.L.M.R) and MRFF Investigator Grant MRF1173594 (to I.W.D.). The contents of the published materials are solely the responsibility of the administering institution, a participating institution or individual authors, and they do not reflect the views of the NHMRC or MRFF. We thank Xavier Godron (DNA Script), Nadège Tardieu (DNA Script), Alexandre Evans (DNA Script) and Fayza Cherradou (DNA Script) for assistance in the production of enzymatically synthesised DNA oligos using the SYNTAX System. We also thank Jeff Jeddeloh (DNA Script), Marky Appel (DNA Script), Bailey Schmidt (DNA Script) and Randy Dyer (DNA Script) for their assistance in experimental design and manuscript preparation.

## Author contributions

H.M.G., A.L.M.R., N.S.S., E.M. and T.R.M. conceived the project and devised the experiments. B.S.M., N.S.S. and I.S. prepared library adaptors, samples and sequenced NGS libraries and conducted laboratory experiments. S.E.Y., H.M.G., T.R.M., S.K.K. I.W.D. and T.W. performed data analysis. S.E.Y., H.M.G., S.K.K. and T.R.M. prepared the manuscript, with support from all co-authors.

## Competing interests

T.R.M. and the Garvan Institute have submitted a patent application to the US patent office pertaining to the design and use of control library adaptors (CAPTORS). Provisional application: 2020900401; 2020. The remaining authors declare no competing interests.
