## [Peer Review File · Nature Communications]

REVIEWER COMMENTS

Reviewer #1 (Remarks to the Author):

The manuscript "Library adaptors with integrated reference controls improves accuracy and reliability of sequencing." by Helen M. Gunter and colleagues describes a new method to label fragments in DNA libraries with so called "CAPTORS" to improve sequencing results.

Overall, the manuscript is well and comprehensively written. Nevertheless, a thorough review of the text is needed. There are several formatting issues I could identify (see below) but I would not be surprised if there are more that skipped my eye.

The methods used and data presented are appropriate to support the claims and findings of the manuscript. The benefit of using CAPTORS, when sequencing long molecules using third generation technologies, is clearly and convincingly presented. Unfortunately, the claim, that the method is also beneficial for the sequencing of short molecules using second generation sequencing, is not substantiated with any data.

Therefore, it would be nice to see the performance of CAPTORS for short read libraries and to get an estimate if the benefits outweigh the costs (both actual \$ and loss of information due to shorter reads).

Typos and other technical issues in the manuscript:

Line 49: cannot ****be**** used ...

Line 132: ... performance of difference sequencing ... should either be: ... performance of differences in sequencing ... or ... performance of different sequencing ...

Line 136: ... CAPTOR libraries ****to**** compare ...

Header in figure S2a should not be >4/5nt homopolymers since it is (based on the legend) homopolymers of exactly 4/5 nts.

Sentence of lines 53, 54 and 55 incomprehensible.

CAPTORS are sometimes written with a capital S in the manuscript but a small s in the figures and figure legends.

In figure S4e the legend says for the last window 24-72 hours, the plot says 24-74 hours.

In figure S4b what does the R9 / R10 on the bottom stand for? The R9.4 and R10.3 pores respectively?

Figure 2c: For me it would be more intuitive to have a "cold" color for low error rates and a "warm" color for high ones.

Line 142: figure 5c should be figure S5c (?).

Figure 3: subfigures c and d seem to be mixed up (figure vs. legend).

Figure S7c: hu****m****an in header instead of huan.

Figure S7c: other than mentioned in the manuscript (line 222) there is no reference to GAPDH in the figure.

Figure S8a: BRCAptors (in header) vs. BRCAPTOR(s) in legend and plot.

Figure 5: Three different ways to write BRCAPTORS (BRCAPTORS vs. BRCAPTORs vs. BRCAptors).

Line 259: ... median error rate $\sim\sim^{**}of^{**}\sim\sim$ was reduced ...

Line 361 ... Where a 6-mer was present $**in^{**}$ more than one ...

Reviewer #2 (Remarks to the Author):

In this paper, the authors detail CAPTORS, which are adaptors that can be used in long-read sequencing. Throughout the manuscript they use CAPTORS for several purposes including measuring base-call error biases in Nanopore sequencing, measuring pore performance, creating a quantification reference ladder, as a normalization factor to reduce batch effects in fold-change, and to improve error correction for sequencing-based cancer diagnosis. CAPTORS can be used to generate useful insights into long-read sequencing libraries, and they provide a tool to compare sequencing platforms while avoiding some of the limitations of traditional spike-in controls.

The data and results which are presented in this manuscript are of high quality and provide useful insights specifically regarding the Nanopore sequencing platform, such as the 6-mer error rate biases and the limits of confidence for gene quantification relative to the number of reads sequenced. The manuscript is well written, clearly structured, and the analysis seems to be performed robustly. While the results are presented clearly, the application of CAPTORS for general usage is less clear.

Major

1. The authors write that CAPTORS can be used for NGS in general, but results are focused on Nanopore sequencing. Given that the adapters are 90nt and that Illumina short reads range from 50-300bp, it is not immediately clear if they will work well for short reads. I suggest that the authors either show data on other sequencing technologies, or rephrase their claims to be specific to the Nanopore technology (or long reads).

2. The manuscript focuses on the future general application of CAPTORS, however the majority of the use-cases seem to be technology-specific, and the same results might be expected to what is presented in this manuscript. I would suggest to emphasise the usefulness of the results that are shown (with CAPTORS being the method by which it was obtained) as the use case for CAPTORS in many of the scenarios beyond what is shown here is not entirely clear. For example:

1. Using CAPTORS the authors show that base calling errors rates are similar to the error rates of the first sequenced adapter region. Now that this is known as reported by this manuscript, the use case for CAPTORS for this purpose by general users is not immediately clear. Can the authors speculate in the discussion if this can be applied to the observed error rates of non-CAPTOR adapters for the purpose of measuring pore error rates? What are additional use cases for this scenario?

2. In the manuscript it is shown that $\sim 5 \times 10^4$ reads are required to achieve reliable quantification across the full range of CAPTORS. This relationship between the number of sequenced reads vs the threshold of quantification is useful and applicable to future experiments. I would expect that this finding has a lot of generality and should be applicable to other experiments without needing to use

CAPTORS in the sequencing run. This conflicts with the presented argument that CAPTORS could/should be continued to be used to ensure minimal sequencing thresholds. This claim could be adjusted or backed with more evidence, or the applications could again be mentioned in the discussion.

3. A key motivation is that CAPTORS overcome the limitation of spike in controls to depend on the spike-in concentration. A comparison of CAPTORS with spike-ins could help to demonstrate this and provide a strong argument to use CAPTORS.

4. Many of the applications that are shown in the manuscript require specialized analysis scripts that provide a substantial barrier for other users. The authors could provide the code to perform such analysis for users of CAPTORS which would reduce the barrier for applications.

5. At the end of the manuscript it is argued that BRCAPTORS, or CAPTORS in general, could help identify clinically relevant mutations.

1. In their example they show how their method reduces the error rate for this reason, presumably to make real changes in the sequence stand out stronger. Could the authors make it clearer how the error correction impacts identification of real mutations by including sites where the sequence does not match the reference?

2. The authors designed BRCAPTORS to measure the expected error rate of a genomic region with some bases shown to have higher error rates. Do the kmers which show higher error rates also match with what was measured earlier on error rates of kmers, and can the authors show this? If they are different, could the authors discuss why there would be a difference in these error rates, and if there is not a difference, could the authors explain the advantages of designing new adapters instead of normalizing using the k-mer error rates seen in the regular CAPTORS?

Minor

1. Fig. 1c shows that there is variance in the sequencing accuracy of the 6-mers. In the section "Measuring individual pore performances using CAPTORS" it is claimed that the sequencing accuracy of the CAPTOR matches that of the subsequent read. Could the authors clarify if the differences in sequencing accuracy of the varying 6-mer region is included in this calculation and thereby affects the accuracy of the rest of the read? Or is only the 3' end of the CAPTOR which is the same in all CAPTORS used to measure the error frequency.

2. Currently the authors only provide how many reads are required to be able to quantify the full range of CAPTORS. This number is specific to the CAPTOR pool and can not easily be translated to how many reads would be needed for a real sample. One possible solution is stating that $\sim 5 \times 10^4$ reads are requested to detect an X% fraction of a sample at a Y level of certainty, or showing a figure that details this relationship.

3. Throughout Figure 3 and the section on using CAPTORS to measure quantitative sequencing the expected concentrations of CAPTORS are used and then correlated with read count. This is standard when analysing spike-ins. However, unlike spike-ins, I presume the concentration of the adapter mastermix (above saturating levels of the RNA) should not affect the observed read count. As it is the relative concentration levels between the adapters is what is important, this could be rewritten for clarity

4. Fig S7.b - shouldn't the LOQ dashed line be horizontal and match 7.1 reads?

Additional comments

Below are small notes for the benefit of the authors that are not included in the evaluation of the manuscript.

58 - These analysis -> These analyses

134 - proteins -> protein

Figure 3 figure legends for c and d are mixed up

Figure 4 a No normalised -> Not normalized/no normalization

184 - points to Fig. S7a but the figure doesn't relate to the text

No explanation what the blue and green means in fig S7.c, is it the same as fig S7.b and that green is below LOQ?

What do the blue dots on the 0 axis line represent in figure 4.a?

Error bar is hard to see for the not normalized plot of 4.a

205 - its' -> its

259 - "finding the median error rate of was reduced from" (remove of?)

285 - These quantitative metrics are need to measure (are needed?)

Inconsistent use of CAPTORS vs CAPTORs

Reviewer #3 (Remarks to the Author):

"Library adaptors with integrated reference controls improves accuracy and reliability of sequencing." by Gunter et al. describes the development of library adaptors specifically designed for error-correcting ONT seq data. The authors show how the library adaptors can be used to improve variant calling accuracy for clinical diagnostics as well as quantitative metagenomics and transcriptome studies. A primary limitation of long-read pore-based sequencing methods is the base-level accuracy and these library adaptors complement the continual advancements in basecalling algorithms. A primary limitation of the study is insufficient methods for reproducing the data analysis presented in the paper. Ideally, a set of scripts used to perform the analysis would be provided as supplemental or via a public code repository like GitHub. Additionally, as ONT basecalling algorithms are rapidly evolving the manuscript would benefit from a demonstration of how the proposed error correction

methods work with different base callers and/or deep learning models.

Major Revisions

- * Additional analyses of how the library adaptors work with different basecallers. I would particularly like to see how error correction performance compares between libraries called using guppy v3, v4, and v5. How does the overall read accuracy compare? Do differences in the measured 6-mer error profile match the developer's statement about differences in model accuracy between versions?
- * Please make the commands and or code used to analyze the basecalled sequencing data available for readers to verify the presented results and more easily use the CAPTORs in their own work.
- * Please expand on the applicability of these CAPTORs to other long-read sequencing technologies and short-read methods. How would the design considerations differ? What is the potential utility, if any, in making a universal / cross-platform set of library adaptors?
- * Please describe the methods used for validation of CAPTORs synthesis - testing oligo accuracy and purity. If relevant, please state how sequence synthesis errors or oligo impurities would impact the error correction.
- * A diagram of quantitative mixture generation would help readers more easily understand the study design and utility of the mixtures for normalization.

Minor Revisions

- * Text edits: Introduction line 54 - "class of library adaptors the encode" should be "class of library adaptors that encode". Methods line 365 - Missing period in "header details To evaluate" should be "header details. To evaluate"
- * Methods - specific guppy version and model used for basecalling.
- * Fig2C error rate scale bar counter-intuitive, I would have assumed red indicates higher error rates.
- * Fig2A and 2B smoothing over points might help show the difference and lack of difference in error rates between types of errors and replicates. Kmer rank correlations between replicates might provide insight into systematic errors.

Reviewed by Nathan Olson

Reviewer #4 (Remarks to the Author):

The manuscript describes a novel NGS adapter design that incorporates a 30bp variable region that encodes various 6-mer sequence-space contents to allow for various error checking and quantification approaches. The authors demonstrate the application of the CAPTOR adapters for these applications on the Oxford Nanopore platform. The manuscript is clear and the experiments and statistical analysis are well designed to demonstrate these various applications. Overall, I think the CAPTOR adapters do work as described and may be useful for various applications, particularly for quantitation.

The major critiques I have are as follows:

1. In my opinion the CAPTOR application is only going to be useful for the Oxford platform which is both long-read and error prone. For the Illumina short-read platform with total read length typically of 300bp (2x150) the CAPTOR would use too much of the total read length and as such take away from the DNA or RNA sequencing of interest. For both the Illumina and the PacBio HiFi reads the sequence quality is very high and doesn't require error checking using adapter sequence. In the abstract, the end of the discussion, and in several places in the text they explicitly say or imply that CAPTORs would be useful to any NGS platform, but I can only see them really being of value with Oxford Nanopore for error checking.
2. The authors do show that the CAPTORs correctly reflect the 6-mer error profile of the Oxford Nanopore platform, however, they do not demonstrate that they are useful in correcting these errors or filtering reads for general application like de novo assembly. The specific application for error checking for cancer gene mutant profiling could be of interest, though the scope is going to be limited

as there is a lot of upfront work required for each gene to be profiled. For a small number of highly informative genes (i.e. BRACA) this could be of value of course but for larger panels spike-ins of genes of interest may be more straightforward and cost-effective.

3. The authors mention that the CAPTOR could help with the Nanopore adaptive sequencing strategies. This seems like a potentially powerful use when a CAPTOR is being misread that the DNA molecule could be ejected from the pore to free it up for a higher-quality read. This definitely seems like an interesting possibility, but I'm not sure the 30bp would be sufficient to accurately model 'bad reads' as I believe these types of approaches would look at a substantially longer region before making the decision to reject or keep. If this is a potential limitation some additional estimates of how long the CAPTOR would need to be for this application would be useful just to have a sense of what would CAPTOR-adaptive sequencing look like in the real-world

4. Overall, I think the quantification applications are more compelling than the error checking applications. The authors do a good job of showing two common applications that would benefit from the use of CAPTOR. The use of CAPTOR to support Oxford Nanopore RNA quantification which is not common yet seems like a powerful application of this technologies. The metagenome normalization is also compelling. High molecular weight spike-ins at different concentrations could be used for the same application, but the CAPTORs definitely provide a simpler process and can avoid overwhelming the target sample which can happen with spike-ins.

The authors would like to thank the reviewers for their time, effort and constructive feedback that has improved the manuscript.

REVIEWER COMMENTS

Reviewer #1 (Remarks to the Author):

The manuscript "Library adaptors with integrated reference controls improves accuracy and reliability of sequencing." by Helen M. Gunter and colleagues describes a new method to label fragments in DNA libraries with so-called "CAPTORS" to improve sequencing results.

Overall, the manuscript is well and comprehensively written. Nevertheless, a thorough review of the text is needed. There are several formatting issues I could identify (see below), but I would not be surprised if there are more than skipped my eye.

The methods used and the data presented are appropriate to support the claims and findings of the manuscript. The benefit of using CAPTORS, when sequencing long molecules using third-generation technologies, is clearly and convincingly presented. Unfortunately, the claim, that the method is also beneficial for the sequencing of short molecules using second-generation sequencing, is not substantiated with any data.

Therefore, it would be nice to see the performance of CAPTORS for short read libraries and to get an estimate of the benefits outweigh the costs (both actual \$ and loss of information due to shorter reads).

In response to the Reviewer's request, we have developed proof-of-principle CAPTORS to show their performance for short-read Illumina sequencing. Briefly, we integrated control elements into Y-adaptors used in Illumina sequencing. These CAPTORS were then titrated across a quantitative range to formulate a mixture that was used for the preparation of DNA libraries for Illumina sequencing. We described a preliminary, proof-of-principle experiment in the manuscript (see below).

As the Reviewer correctly points out, the short-reads limit the size of the control elements that can be incorporated, and therefore limits their utility for measuring sequencing accuracy. Nevertheless, we found that the CAPTORS are still useful as quantitative reference standards for Illumina sequencing. Please note that because they are incorporated into the control elements within the Y-adaptors (essentially replacing other sequences), they do not incur any cost in loss of information.

“Using CAPTORS with short-read sequencing.

Library adaptors are used during the preparation of sample DNA for short-read (Illumina) sequencing, where they are required to attach the DNA to the flowcell substrate and initiate the sequencing-by-synthesis reaction. We next considered whether we could similarly develop CAPTORS that are compatible with Illumina sequencing. To provide a proof-of-principle, we first designed four pairs of CAPTORS, each containing a 12nt control elements (**Fig. S11a**). The CAPTOR elements were necessarily shorter than our ONT adaptors, and limited to minimise complementarity to other sequences within the adaptor (less than five contiguous nucleotides to the P5, P7 sequences or index sequences). The control elements also encoded different nucleotides at corresponding sites to ensure optimum cluster discrimination at each sequencing cycle. Each CAPTOR was paired and hybridised to form Y-shaped adaptor that was then combined at different concentrations to formulate a custom adaptor mixture for Illumina sequencing (see **Methods**).

These adaptor sequences were used during the preparation of three replicate libraries from a reference sample comprising a synthetic mock microbial community. The resulting library was sequenced, and control elements were identified at the 5' start of sequenced reads in the output library (see **Methods**). By comparing the observed count of CAPTORS elements to their expected relative concentration, we assembled a quantitative reference ladder (**Fig. S11b**). This plot indicates the library's quantitative accuracy ($R^2=0.9888$) and linearity. By comparing CAPTOR counts between replicates, we could also estimate the technical variation. We propose that combining these quantitative CAPTORS with unique molecular identifiers (UMI) would further improve the quantitative analysis and normalisation of short-read (Illumina) library preparation.”

We also included the following description in Methods:

“Design and analysis of CAPTORS for short-read Illumina sequencing.

We first designed 4 CAPTOR Y-adaptor sequences (A-D) used in Illumina sequencing. Each CAPTOR included a control element comprising 12nt, that were randomly generated but filtered to exclude low-complexity or repeat sequences. Control sequences also had minimal complementarity (less than 5 contiguous nucleotides) to other sequences within the adaptor (including P5, P7 sequences). The control elements also encoded different nucleotides at corresponding sites to ensure optimum cluster discrimination during each sequencing cycle. Each adaptor was paired to a corresponding adaptor and hybridised by cooling from 68°C over 20 minutes to enable annealing between each pair to form Y-shaped adaptors. The polynucleotide Y-adaptors were then combined in the following relative amounts (A = 10%; B = 20%; C = 30% and D = 40%) to formulate a mixture of adaptor sequences (using Eppendorf Emotion).

We then used the CAPTORS mixture to prepare libraries from a test sample using the Illumina TruSeq Library Preparation Kit (according to the manufacturer's instructions, except for the replacement of standard library adaptors with CAPTOR mixture). The test sample comprised a synthetic mock microbial community (Mock Community A). This formed a mixture of synthetic DNA sequences (known as sequins) that mirrored a range of microbial genomes (Hardwick et al., 2019). The prepared library was validated using the Agilent 2100 Bioanalyzer and then sequenced (Illumina® NextSeq) as per the manufacturer's instructions. The control element was then identified at the 5' start of sequenced reads in the output library (.FASTQ) using *cutadapt* (Martin et al., 2011; cutadapt 1.8.1 with Python 2.7.6), with the modification of providing our custom adaptor sequences (-g option) that comprised the calibration sequences."

We also included an additional **Supplementary Figure 11**.

Typos and other technical issues in the manuscript:

Line 49: cannot ****be**** used ...

Line 132: ... performance of difference sequencing ... should either be: ... performance of differences in sequencing ... or ... performance of different sequencing ...

Line 136: ... CAPTOR libraries ****to**** compare ...

Header in figure S2a should not be >4/5nt homopolymers since it is (based on the legend) homopolymers of exactly 4/5 nts.

Sentence of lines 53, 54 and 55 incomprehensible.

CAPTORS are sometimes written with a capital S in the manuscript but a small s in the figures and figure legends.

In figure S4e the legend says for the last window 24-72 hours, the plot says 24-74 hours.

In figure S4b what does the R9 / R10 on the bottom stand for? The R9.4 and R10.3 pores respectively?

Figure 2c: For me it would be more intuitive to have a "cold" color for low error rates and a "warm" color for high ones.

Line 142: figure 5c should be figure S5c (?).

Figure 3: subfigures c and d seem to be mixed up (figure vs. legend).

Figure S7c: hu****m****an in header instead of huan.

Figure S7c: other than mentioned in the manuscript (line 222) there is no reference to GAPDH in the figure.

Figure S8a: BRCAptors (in header) vs. BRCAPTOR(s) in legend and plot.

Figure 5: Three different ways to write BRCAPTORS (BRCAPTORS vs. BRCAPTORs vs. BRCAptors).

Line 259: ... median error rate **~**of**~** was reduced ...

Line 361 ... Where a 6-mer was present ****in**** more than one ...

We thank the Reviewer for identifying these errors that how now been corrected in the manuscript.

Reviewer #2 (Remarks to the Author):

In this paper, the authors detail CAPTORS, which are adaptors that can be used in long-read sequencing. Throughout the manuscript, they use CAPTORS for several purposes, including measuring base-call error biases in Nanopore sequencing, measuring pore performance, creating a quantification reference ladder, as a normalisation factor to reduce batch effects in fold-change, and to improve error correction for sequencing-based cancer diagnosis. CAPTORS can be used to generate useful insights into long-read sequencing libraries, and they provide a tool to compare sequencing platforms while avoiding some of the limitations of traditional spike-in controls.

The data and results which are presented in this manuscript are of high quality and provide useful insights specifically regarding the Nanopore sequencing platform, such as the 6-mer error rate biases and the limits of confidence for gene quantification relative to the number of reads sequenced. The manuscript is well written, clearly structured, and the analysis seems to be performed robustly. While the results are presented clearly, the application of CAPTORS for general usage is less clear.

Major

1. The authors write that CAPTORS can be used for NGS in general, but results are focused on Nanopore sequencing. Given that the adapters are 90nt and that Illumina short reads range from 50-300bp, it is not immediately clear if they will work well for short reads. I suggest that the authors either show data on other sequencing technologies, or rephrase their claims to be specific to the Nanopore technology (or long reads).

Our study focused on the design of CAPTORS for nanopore sequencing, which provides a greater length of sequencing reads and therefore greater opportunities to design CAPTOR features. However, CAPTORS can also be used with short-read Illumina sequencing, however, as noted by the Reviewer, the benefits are more limited.

To show the compatibility of CAPTORS with Illumina sequencing, we performed a proof-of-principle experiment that is summarised "*Using CAPTORS with short-read sequencing*" and **Supplementary Figure 11** (see Response 1). However, we also include the following discussion of the limitations of CAPTORS with short-read sequencing in the discussion as follows:

"We provided a proof-of-principle demonstration that CAPTORS are compatible with short-read Illumina sequencing. Due to the short read length, the control elements are necessarily short and do not encode extended reference sequences, required to provide a comprehensive analysis of sequencing accuracy. Nevertheless, the CAPTORS can provide quantitative reference ladders to measure the sensitivity and accuracy of short-read sequencing libraries."

In addition, we have also clarified throughout the manuscript when we are specifically referring to the use and benefits of CAPTORS for either long- or short-read sequencing (see track changes in manuscript).

2. The manuscript focuses on the future general application of CAPTORS. However, the majority of the use-cases seem to be technology-specific, and the same results might be expected to what is presented in this manuscript. I would suggest emphasising the usefulness of the results that are shown (with CAPTORS being the method by which it was obtained) as the use case for CAPTORS in many of the scenarios beyond what is shown here is not entirely clear. For example:

Using CAPTORS, the authors show that base-calling errors rates are similar to the error rates of the first sequenced adapter region. Now that this is known as reported by this manuscript, the use case for CAPTORS for this purpose by general users is not immediately clear. Can the authors speculate in the discussion if this can be applied to the observed error rates of non-CAPTOR adapters for the purpose of measuring pore error rates? What are additional use cases for this scenario?

While error rates can be determined from the known sequences of standard (non-CAPTOR) library adaptors, these include only a narrow representation of sequences and do not provide a comprehensive measure of sequencing accuracy. We have described this advantage of CAPTORS in the discussion as follows:

"The CAPTORS comprehensively represent different k-mers or specific gene sequences of interest (that cannot be otherwise determined from standard library adaptors)."

The Reviewer also asks for additional uses cases for CAPTORS (after being used to measure sequencing performance). However, the sequencing accuracy is impacted by different reagents, base-calling software and pore versions (amongst other variables), and we, therefore, anticipate users will benefit from regularly measuring sequencing accuracy. This ability to routinely measure sequencing performance can also provide surveillance of performance, quality control, rapid troubleshooting, and benchmarking of different methods, reagents or instruments. For example, within this study, we show how the CAPTORS can be used to compare the performance of alternative R9 and R10 nanopore versions. The use cases are anticipated to be particularly important as nanopore sequencing is translated into clinical diagnosis. We have highlighted these routine use cases in the discussion as follows:

“Like other reference standards, CAPTORS can routinely measure sequencing performance and quality control, enable rapid troubleshooting, and benchmark different methods, reagents or instruments. Routine use of CAPTORS, which can be seamlessly incorporated into the NGS workflow, will measure performance and inform operational decisions.”

3. In the manuscript it is shown that $\sim 5 \times 10^4$ reads are required to achieve reliable quantification across the full range of CAPTORS. This relationship between the number of sequenced reads vs the threshold of quantification is useful and applicable to future experiments. I would expect that this finding has a lot of generality and should be applicable to other experiments without needing to use CAPTORS in the sequencing run. This conflicts with the presented argument that CAPTORS could/should be continued to be used to ensure minimal sequencing thresholds. This claim could be adjusted or backed with more evidence, or the applications could again be mentioned in the discussion.

We agree with the Reviewer that the ability of CAPTORS to empirically determine the minimum sequencing depth threshold is useful, and once determined, this threshold can be applied to other experiments. However, the minimum sequencing depth threshold will depend on the user's experiment, method and aims. For example, the minimum threshold needed to diagnose a somatic mutation accurately will be much higher than the minimum depth threshold needed to measure the expression of highly expressed genes. Furthermore, this threshold may need to be recalculated when using different reagents or analyses. We would also highlight that because CAPTORS can be seamlessly integrated within library preparation at no additional cost, there is no cost incurred by using CAPTORS routinely. Therefore, we have described this motivation for routinely using CAPTORS in the results as follows:

“This demonstrates how ongoing real-time analysis of the CAPTORS could be used to ensure minimal sequencing thresholds are attained according to the desired level of accuracy and sensitivity. This minimum threshold may vary between experiments, and will depend on several factors, including the experimental aims, desired sensitivity, and the particular analysis performed^{24–26}.”

We also disagree that there are no further benefits for using CAPTORS once minimum thresholds have been determined. CAPTORS can enable normalisation between different libraries prepared with alternative kits or laboratories. This ability to normalise between samples (described in detail in the "**Normalisation of metagenome samples with CAPTORS**" section) can mitigate batch-effects and improve the detection of differences between samples. However, this requires the routine use of CAPTORS in each sample.

4. A key motivation is that CAPTORS overcome the limitation of spike-in controls to depend on the spike-in concentration. A comparison of CAPTORS with spike-ins could help to demonstrate this and provide a strong argument to use CAPTORS.

To validate our CAPTORS, we compared their performance (to measure both sequencing and quantitative accuracy) to microbial spike-ins used for metagenomics. These spike-in controls provide a useful reference sample with known quantities against which to compare CAPTORS, and have been previously described in detail (4). Our analysis shows that the CAPTORS perform equivalently to the spike-in controls.

However, the Reviewer is correct that a key disadvantage of spike-ins is that they must be added at the correct concentrations and must often be bioinformatically subsampled to the correct depth prior to analysis. This can be particularly problematic for low concentration or degraded samples. To highlight this limitation of spike-ins relative to CAPTORS, we have included the following statements:

“Although synthetic spike-ins have the advantage of measuring internal library variation, they must be precisely added to a sample during library preparation, must be bioinformatically calibrated, and risk overwhelming low input or degraded samples.”

5. Many of the applications that are shown in the manuscript require specialised analysis scripts that provide a substantial barrier for other users. The authors could provide the code to perform such analysis for users of CAPTORS which would reduce the barrier for applications.

The commands used to analyse the sequencing data solely employ publicly available software tools, including *MiniMap2* and *psysamstats*. Scripts used for analysis of CAPTORS can be accessed via: <https://github.com/mercertim/Captors> (as indicated in DATA AVAILABILITY section).

6. At the end of the manuscript, it is argued that BRCAPTORS, or CAPTORS in general, could help identify clinically relevant mutations. In their example they show how their method reduces the error rate for this reason, presumably to make real changes in the sequence stand out stronger. Could the authors make it clearer how the error correction impacts the identification of real mutations by including sites where the sequence does not match the reference?

The BRCAPTORS provide an internal reference gene sequence against which to compare sequencing of matched genome regions in the accompanying patient sample. Specifically, the BRCAPTOR provides a per-nucleotide background error rate against which sequencing of the BRCA gene in the patient sample can be compared. This ‘error correction’ can prevent the false-positive diagnosis of ‘mutations’ that result from technical sequencing errors and indicate the assay’s accuracy limit (i.e. the minimum allele frequency below which it is impossible to distinguish mutations from sequencing errors).

The Reviewer asks whether BRCAPTORS could include mutations that do not match the reference and could be used to measure true-positive diagnosis. This is certainly a possibility; however, this would not markedly impact the detection of true-positive mutations (and therefore sensitivity). Therefore, we do not include mutations in our BRCAPTORS as this sequence adds little utility and then can not be used as a ‘background’ against which to normalise error correction in patient samples.

7. The authors designed BRCAPTORS to measure the expected error rate of a genomic region with some bases shown to have higher error rates. Do the kmers which show higher error rates also match with what was measured earlier on error rates of kmers, and can the authors show this? If they are different, could the authors discuss why there would be a difference in these error rates, and if there is not a difference, could the authors explain the advantages of designing new adapters instead of normalising using the k-mer error rates seen in the regular CAPTORS?

The advantage of designing new CAPTORS that represent specific gene sequences (such as *BRCA* exons) is that they provide a matched background sequence against which to interpret sequencing performance. The representation of a genetic sequence of clinical interest enables empirical determination of the ‘background’ sequencing error rate, which can be used to normalise the error profile in matched regions in the patient sample.

By contrast, measuring the sequencing accuracy of k-mers with regular CAPTORS provides a useful, detailed and comprehensive sequencing error profile. However, as the Reviewer notes in the response below, there is some variation in the k-mer sequencing accuracy, and this error profile is not sufficiently robust for normalisation of errors in specific sequences of clinical interest (such as *BRCA* exons). This has been clarified in the following sentence in the results:

“This proof-of-principle experiment demonstrates how CAPTORS containing clinically important sequences can provide internal controls to guide error-correction tools and improve the interpretation and accuracy of ONT sequencing data during clinical diagnosis³⁶. However, while this approach can include genes of diagnostic importance, it is limited to smaller gene panels, and standard spike-ins may be more suitable for representing larger numbers of genes.”

1. Fig. 1c shows that there is variance in the sequencing accuracy of the 6-mers. In the section “Measuring individual pore performances using CAPTORS” it is claimed that the sequencing accuracy of the CAPTOR matches that of the subsequent read. Could the authors clarify if the differences in sequencing accuracy of the varying 6-

mer region are included in this calculation and thereby affects the accuracy of the rest of the read? Or is only the 3' end of the CAPTOR which is the same in all CAPTORS used to measure the error frequency.

As described in the study, our comparison of the mean accuracy of CAPTORS is similar to the associated read. However, the Reviewer is correct in that individual 6-mers show substantial variation in accuracy (see response above), and we, therefore, expect to see some variation between CAPTOR and read accuracy. Although this variation is averaged out across the 25 6-mers present in the variable region of each CAPTOR, this may bias results. Therefore, we have amended and clarified this in the manuscript as follows:

“We found that mean CAPTOR sequencing accuracy matches the mean sequencing accuracy of the adjacent microbial DNA sequence (Fig. S5a).”

2. Currently the authors only provide how many reads are required to be able to quantify the full range of CAPTORS. This number is specific to the CAPTOR pool and can not easily be translated to how many reads would be needed for a real sample. One possible solution is stating that $\sim 5 \times 10^4$ reads are requested to detect an X% fraction of a sample at a Y level of certainty, or showing a figure that details this relationship.

The number of reads needed for accurate quantification is equal between the CAPTORS and the sample. This is because each sample DNA fragment is adjoined by two flanking CAPTORS, and therefore the depth required for CAPTOR sequencing is matched to the depth of the accompanying sample. This is a major advantage of CAPTORS compared to spike-in controls, which must be carefully added to the sample and bioinformatically subsampled to provide a relevant quantitative range for the accompanying sample. To provide a clear example of this, as requested by the Reviewer, we show that >5000 -fold coverage is required to detect CAPTORS at 0.78% concentration. This has been included in the results as follows:

“We found a minimum sequencing coverage of $\sim 5 \times 10^4$ reads, which was achieved during the first ~ 2 hours of sequencing, which was required to achieve reliable quantification across the full dynamic range of CAPTORS (to $<1\%$ frequency; Fig. S7d). Below this threshold, we observed increasing quantitative uncertainty illustrated by a wide confidence interval at lower sequencing depth.”

The empirically-determined relationship between sequencing depth and CAPTOR counting/detection (which is a key advantage of CAPTORS) is also illustrated in **Supplementary figure 7d**.

3. Throughout Figure 3 and the section on using CAPTORS to measure quantitative sequencing the expected concentrations of CAPTORS are used and then correlated with read count. This is standard when analysing spike-ins. However, unlike spike-ins, I presume the concentration of the adapter mastermix (above saturating levels of the RNA) should not affect the observed read count. As it is the relative concentration levels between the adapters is what is important, this could be rewritten for clarity

We thank the Reviewer for the suggestion that has now been clarified throughout the main text and **Figure 3**. For example:

“The variable CAPTOR sequences were then retrieved from each read, counted and compared to the expected CAPTOR concentration to generate a staggered reference ladder that can measure quantitative library features²² (see Methods).”

4. Fig S7.b - shouldn't the LOQ dashed line be horizontal and match 7.1 reads?

The limit of quantification (LOQ) indicates the lower limit of reliable quantification. This can be expressed as either the read count (as suggested by the Reviewer) or the relative concentration. We agree with the Reviewer that expressing LOQ as read count can be more intuitive, and **Supplementary Figure 7b** has been revised according to the Reviewer's request.

Additional comments

Below are small notes for the benefit of the authors that are not included in the evaluation of the manuscript.

58 - These analysis -> These analyses

134 - proteins -> protein

Figure 3 figure legends for c and d are mixed up

Figure 4 a No normalised -> Not normalized/no normalization

184 - points to Fig. S7a but the figure doesn't relate to the text

No explanation what the blue and green means in fig S7.c, is it the same as fig S7.b and that green is below LOQ?

What do the blue dots on the 0 axis line represent in figure 4.a?

Error bar is hard to see for the not normalised plot of 4.a

205 - its' -> its

259 - "finding the median error rate of was reduced from" (remove of?)

285 - These quantitative metrics are needed to measure (are needed?)

Inconsistent use of CAPTORS vs CAPTORs

We thank the Reviewer for identifying these errors that how now been corrected in the manuscript.

Reviewer #3 (Remarks to the Author):

“Library adaptors with integrated reference controls improves accuracy and reliability of sequencing.” by Gunter et al. describes the development of library adaptors specifically designed for error-correcting ONT seq data. The authors show how the library adaptors can be used to improve variant calling accuracy for clinical diagnostics as well as quantitative metagenomics and transcriptome studies. A primary limitation of long-read pore-based sequencing methods is the base-level accuracy and these library adaptors complement the continual advancements in basecalling algorithms. A primary limitation of the study is insufficient methods for reproducing the data analysis presented in the paper. Ideally, a set of scripts used to perform the analysis would be provided as supplemental or via a public code repository like GitHub. Additionally, as ONT basecalling algorithms are rapidly evolving the manuscript would benefit from a demonstration of how the proposed error correction methods work with different base callers and/or deep learning models.

We have included all the scripts used to analyse the CAPTORs in **Supplementary Methods**. For the analysis, we use publicly-available software tools, and have not developed a dedicated software package for these CAPTORs as the analysis is relatively straightforward.

Major Revisions

1. Additional analyses of how the library adaptors work with different basecallers. I would particularly like to see how error correction performance compares between libraries called using guppy v3, v4, and v5. How does the overall read accuracy compare? Do differences in the measured 6-mer error profile match the developer's statement about differences in model accuracy between versions?

We agree that the CAPTORs would be an effective tool for benchmarking the performance of different guppy base-callers (v3-5). However, whilst CAPTORs can be used to benchmark different base-callers, software and protocols, we have refrained from undertaking extensive benchmarking in this manuscript because we consider a rigorous evaluation of these different variables to be beyond the scope of the study, and there have also been numerous recent studies that have performed a detailed benchmarking (e.g. Wick et al. 2019, *Genome Biology*; Silvestre-Ryan & Holmes 2021, *Genome Biology*). Nevertheless, we do agree that CAPTORs can be easily and routinely used to benchmark different methods, and within the study, we used CAPTORs to compare the quality of data generated by the new R10 pore, vs the R9 pore as follows:

“CAPTORs can also benchmark the performance of different sequencing reagents and methods. We used CAPTORs to evaluate the sequencing accuracy of different nanopore versions. The R10.3 nanopore, which has a longer barrel and a dual reader head, has been developed to enhance the accuracy of homopolymer regions²¹. We used matched CAPTOR libraries to compare the error profile of the R10.3 pore to R9.4 pore performance. As expected, the R10.3 pore exhibited a distinct error profile, with a lower mean error rate (0.037 error/nt) compared to the R9.4 pore (0.045 error/nt), which is largely due to the lower insertion rate for the R10.3 pore (0.021 error/nt, compared to the 0.032 error/nt for R9.4, **Fig. S6a,b**). This distinction in R10.3 pore performance, as measured by CAPTORs, is most notable at low-complexity repeats (R10.3 mean error = 0.048, R9.4 mean error = 0.083, **Fig. S6c**). These empirically-determined sequencing error rates differ from manufacturer's reports²¹ and demonstrate how CAPTORs can measure the sequencing performance of each library, benchmark new chemistries and base-calling algorithms and inform best-practise guidelines to optimise sequencing performance.”

2. Please make the commands and or code used to analyse the basecalled sequencing data available for readers to verify the presented results and more easily use the CAPTORs in their own work.

The commands used to analyse the sequencing data solely employ publicly available software tools, including *MiniMap2* and *psysamstats*. Scripts used for analysis of CAPTORs can be accessed via: <https://github.com/mercertim/Captors> (as indicated in DATA AVAILABILITY section).

3. Please expand on the applicability of these CAPTORs to other long-read sequencing technologies and short-read methods. How would the design considerations differ? What is the potential utility, if any, in making a universal / cross-platform set of library adaptors?

In our study, we focused on the design of CAPTORs for use with long-read Oxford Nanopore sequencing. However, CAPTORs are also compatible with short-read sequencing, although some benefits will be limited. For

example, the shortened length of library adaptors in Illumina sequencing limits the size and information that can be encoded in CAPTORS. Furthermore, Illumina sequencing is more accurate, with less requirement for sequencing error correction. Nevertheless, the quantitative benefits of CAPTORS are still useful for short-read sequencing. We performed a proof-of-principle experiment to show CAPTORS are compatible with Illumina short-read sequencing. This additional experiment is summarised in **“Using CAPTORS with short-read sequencing”** and **Supplementary Figure 11** (see Response 1). We have also included the following statements describing differing design considerations for CAPTORS for short- and long-read sequencing as follows:

“Within this study, we largely designed and built CAPTORS for use with nanopore sequencing, whose long-read and error profile benefits from CAPTORS. However, CAPTORS can also be used with other sequencing platforms. We provided a proof-of-principle demonstration that CAPTORS are compatible with short-read Illumina sequencing. Due to the short read length, the control elements are necessarily short and do not encode extended reference sequences, required to provide a comprehensive analysis of sequencing accuracy. Nevertheless, the CAPTORS can provide quantitative reference ladders to measure the sensitivity and accuracy of short-read sequencing libraries. Furthermore, barcoded adaptors, which are widely used in single-cell and spatial transcriptome sequencing methods, can similarly incorporate quantitative reference control sequences and confer the benefits of CAPTORS to measure single-cell library complexity and inform normalization between individual cells.”

However, given the distinctly different design considerations for both short- and long-read sequencing, we think it will be difficult to design universal adaptors that are compatible with both long- and short-read platforms whilst retaining benefits (for example, the 90nt CAPTORS used for measuring sequencing accuracy in ONT sequencing are too long for use in Illumina sequencing).

4. Please describe the methods used for validation of CAPTORS synthesis - testing oligo accuracy and purity. If relevant, please state how sequence synthesis errors or oligo impurities would impact the error correction.

CAPTOR accuracy and purity were verified during preparation by DNA Script SYNTAX, which reports >99.4% accuracy¹. This is far below our observed ONT sequencing accuracy and is therefore not expected to impact our error correction.

¹ <https://www.builtwithbiology.com/read/dna-script-presents-new-data-demonstrating-the-feasibility-of-the-companys-enzymatic-synthesis-technology>

5. A diagram of quantitative mixture generation would help readers more easily understand the study design and utility of the mixtures for normalisation.

Thanks for the suggestion, and to improve understanding, we have included a schematic figure (**Supplementary Figure 1**) indicating the generation of the quantitative mixture and experimental design.

Minor Revisions

* Text edits: Introduction line 54 - “class of library adaptors the encode” should be “class of library adaptors that encode”. Methods line 365 - Missing period in “header details To evaluate” should be “header details. To evaluate”

* Methods - specific guppy version and model used for basecalling.

* Fig2C error rate scale bar counter-intuitive, I would have assumed red indicates higher error rates.

* Fig2A and 2B smoothing over points might help show the difference and lack of difference in error rates between types of errors and replicates. Kmer rank correlations between replicates might provide insight into systematic errors.

We thank the Reviewer for identifying these errors that have been corrected in the manuscript.

Reviewer #4 (Remarks to the Author):

The manuscript describes a novel NGS adapter design that incorporates a 30bp variable region that encodes various 6-mer sequence-space contents to allow for various error checking and quantification approaches. The authors demonstrate the application of the CAPTOR adapters for these applications on the Oxford Nanopore platform. The manuscript is clear and the experiments and statistical analysis are well designed to demonstrate these various applications. Overall, I think the CAPTOR adapters do work as described and may be useful for various applications, particularly for quantitation.

The major critiques I have are as follows:

1. In my opinion the CAPTOR application is only going to be useful for the Oxford platform which is both long-read and error-prone. For the Illumina short-read platform with total read length typically of 300bp (2x150) the CAPTOR would use too much of the total read length and as such take away from the DNA or RNA sequencing of interest. For both the Illumina and the PacBio HiFi reads the sequence quality is very high and doesn't require error checking using adapter sequence. In the abstract, the end of the discussion, and in several places in the text they explicitly say or imply that CAPTORS would be useful to any NGS platform, but I can only see them really being of value with Oxford Nanopore for error checking.

We agree that the benefits conferred by CAPTORS are most useful for Oxford Nanopore sequencing, which has long-reads and is error-prone. The ability of CAPTORS to measure and improve sequencing error is more limited for the Illumina sequencing platform that has short-reads (with short library adaptors) and already has a high sequencing accuracy. Nevertheless, the quantitative benefits of CAPTORS, including their use in measuring sensitivity and quantitative accuracy of libraries, and normalisation between samples, are similarly useful for short-read sequencing.

In response to the Reviewer's suggestion, we have also clarified in the main text that long-read Oxford Nanopore sequencing is particularly suited to CAPTORS. However, we have also included a proof-of-principle experiment to show that CAPTORS are compatible with Illumina short-read sequencing (see section "**Using CAPTORS with short-read sequencing**". and **Supplementary Figure 11**). We have also included the following statement in the discussion:

"Within this study, we largely designed and built CAPTORS for use with nanopore sequencing, whose long-read and error profile benefits from CAPTORS. However, CAPTORS can also be used with other sequencing platforms. We provided a proof-of-principle demonstration that CAPTORS are compatible with short-read Illumina sequencing. Due to the short read length, the control elements are necessarily short and do not encode extended reference sequences, required to provide a comprehensive analysis of sequencing accuracy. Nevertheless, the CAPTORS can provide quantitative reference ladders to measure the sensitivity and accuracy of short-read sequencing libraries. Furthermore, barcoded adaptors, which are widely used in single-cell and spatial transcriptome sequencing methods, can similarly incorporate quantitative reference control sequences and confer the benefits of CAPTORS to measure single-cell library complexity and inform normalization between individual cells."

We have also clarified in the discussion and abstract that CAPTORS are most useful in Oxford Nanopore sequencing.

2. The authors do show that the CAPTORS correctly reflect the 6-mer error profile of the Oxford Nanopore platform, however, they do not demonstrate that they are useful in correcting these errors or filtering reads for general application like de novo assembly. The specific application for error checking for cancer gene mutant profiling could be of interest, though the scope is going to be limited as there is a lot of upfront work required for each gene to be profiled. For a small number of highly informative genes (i.e. BRACA) this could be of value of course but for larger panels spike-ins of genes of interest may be more straightforward and cost-effective.

The CAPTORS can provide a reference sequence to determine background error rates and correct false-positive sequencing errors. However, as noted by the Reviewer, this requires genes of interest to be represented and limits this approach to small gene panels. We have investigated training base-calling algorithms on the CAPTORS, and then applied this to the accompanying DNA sample, however, we did not observe significant improvements in sequencing accuracy and performance. Therefore, in response to the Reviewer's suggestion, we have included the following limitation in the text:

“Although the design of gene-specific CAPTORs is not practical for all genes, this approach is suitable for small panels of selected genes with high diagnostic importance and complex error profiles.”

3. The authors mention that the CAPTOR could help with the Nanopore adaptive sequencing strategies. This seems like a potentially powerful use when a CAPTOR is being misread that the DNA molecule could be ejected from the pore to free it up for a higher-quality read. This definitely seems like an interesting possibility, but I'm not sure the 30bp would be sufficient to accurately model 'bad reads' as I believe these types of approaches would look at a substantially longer region before making the decision to reject or keep. If this is a potential limitation some additional estimates of how long the CAPTOR would need to be for this application would be useful just to have a sense of what would CAPTOR-adaptive sequencing look like in the real-world

Given that CAPTORs are the first part of the DNA that is sequenced, they may be analysed in real-time using 'adaptive sequencing' methods. Indeed, Payne et. al., 2021 recently described the use of 'barcode' aware adaptive sampling that enables the selection of different targets via 5' barcodes, thereby demonstrating this is possible in principle. However, we agree that the 30 bp variable regions of our CAPTORs are likely too short to be well suited for providing an accurate and robust profile of sequence error, and longer adapters that capture the full range of sequencing errors may be required. We indicated this limitation in the discussion as follows:

“Given that CAPTORs are the first part of the read to traverse the nanopore channel and be sequenced, they can provide an immediate measure of sequencing performance. This responsive analysis can be incorporated within 'CAPTOR-aware' adaptive sequencing strategies to provide real-time evaluation of library accuracy and complexity²⁰. However, longer CAPTORs that incorporate more sequence content may be needed.”

Overall, I think the quantification applications are more compelling than the error checking applications. The authors do a good job of showing two common applications that would benefit from the use of CAPTOR. The use of CAPTOR to support Oxford Nanopore RNA quantification, which is not common yet seems like a powerful application of this technology. The metagenome normalisation is also compelling. High molecular weight spike-ins at different concentrations could be used for the same application, but the CAPTORs definitely provide a simpler process and can avoid overwhelming the target sample which can happen with spike-ins.

We thank the Reviewer for their comments and agree that the quantitative dimension provided by CAPTORs is a compelling feature that can improve the accuracy and normalisation of analysis, particularly for metagenomics. We also agree that a major advantage of CAPTORs is they can be seamlessly incorporated into library preparation workflow, enabling their routine use and benefits, without incurring further costs. This allows CAPTORs to confer many of the benefits of reference standards whilst avoiding the requirement to spike-in to samples, which must be performed carefully and can result in overwhelming samples (especially, low-input, degraded or FFPE) and often requires subsequent bioinformatic subsampling. By contrast, CAPTORs are ligated in a constant ratio to the accompanying sample DNA fragments, the quantitative performance of the CAPTORs directly matches the quantitative performance of the accompanying DNA sample without requiring careful calibration.

REVIEWERS' COMMENTS

Reviewer #1 (Remarks to the Author):

I would like to thank the authors to have responded to all my concerns. Especially, I would like to thank for including a proof of principle concerning short-read data and for clarifying the usability and applicability of CAPTORS for this technology.

Reviewer #2 (Remarks to the Author):

In the revised version of their manuscript, the authors have updated the text and methods to clarify comments and provide code for analysing CAPTORS from sequencing data. Overall, most questions were replied with additional explanations and clarifications. The scope of the applications of CAPTORS has been clarified and the title and abstract are now more specific to the results that are presented.

Reviewer #3 (Remarks to the Author):

The authors sufficiently addressed my concerns and I recommend accepting the manuscript for publication. While, I think the manuscript would benefit from an example use case comparison of basecaller accuracy between guppy versions. I agree with the authors that an extensive ONT basecaller comparison is out of scope of this paper. I assume that the raw fast5 files will be made available, likely from SRA under the PRJNA781348 BioProject when the sequencing data is made publicly available so that others can do this analysis.

Reviewer #4 (Remarks to the Author):

The authors additions and modifications to the manuscript have addressed all of my critiques of the original submission. The authors clearly demonstrate several valuable applications of their CAPTOR adapter strategy for both long and short read uses. The addition of the software provides a critical missing piece for other researchers to use their system.

Response to reviewers

REVIEWER COMMENTS

Reviewer #1 (Remarks to the Author):

I would like to thank the authors to have responded to all my concerns. Especially, I would like to thank for including a proof of principle concerning short-read data and for clarifying the usability and applicability of CAPTORS for this technology.

Reviewer #2 (Remarks to the Author):

In the revised version of their manuscript, the authors have updated the text and methods to clarify comments and provide code for analysing CAPTORS from sequencing data. Overall, most questions were replied with additional explanations and clarifications. The scope of the applications of CAPTORS has been clarified and the title and abstract are now more specific to the results that are presented.

Reviewer #3 (Remarks to the Author):

The authors sufficiently addressed my concerns and I recommend accepting the manuscript for publication. While, I think the manuscript would benefit from an example use case comparison of basecaller accuracy between guppy versions. I agree with the authors that an extensive ONT basecaller comparison is out of scope of this paper. I assume that the raw fast5 files will be made available, likely from SRA under the PRJNA781348 BioProject when the sequencing data is made publicly available so that others can do this analysis.

Reviewer #4 (Remarks to the Author):

The authors additions and modifications to the manuscript have addressed all of my critiques of the original submission. The authors clearly demonstrate several valuable applications of their CAPTOR adapter strategy for both long and short read uses. The addition of the software provides a critical missing piece for other researchers to use their system.

We thank the anonymous peer reviewers for their helpful comments. In response to the comment from Reviewer #3 regarding the availability of raw sequencing data, the short-read sequencing data are unfortunately no longer available. As such, discussions of work related to these data were removed, with approval from the handling editors. Instead, we describe the design considerations and utility of custom adaptors in short read sequencing in the discussion.